# Neurogenic decisions require a cell cycle independent function of the CDC25B phosphatase

**Frédéric Bonnet[1†], Angie Molina[1†], Mélanie Roussat[1], Manon Azais[2], Sophie Bel-Vialar[1], Jacques Gautrais[2], Fabienne Pituello[1*], Eric Agius[1*]**

[1]Centre de Biologie du Développement, Centre de Biologie Intégrative, Université de Toulouse, CNRS, UPS, Toulouse, France; [2]Centre de Recherches sur la Cognition Animale, Centre de Biologie Intégrative., Université de Toulouse, CNRS, UPS, Toulouse, France

**Abstract** A fundamental issue in developmental biology and in organ homeostasis is understanding the molecular mechanisms governing the balance between stem cell maintenance and differentiation into a specific lineage. Accumulating data suggest that cell cycle dynamics play a major role in the regulation of this balance. Here we show that the G2/M cell cycle regulator CDC25B phosphatase is required in mammals to finely tune neuronal production in the neural tube. We show that in chick neural progenitors, CDC25B activity favors fast nuclei departure from the apical surface in early G1, stimulates neurogenic divisions and promotes neuronal differentiation. We design a mathematical model showing that within a limited period of time, cell cycle length modifications cannot account for changes in the ratio of the mode of division. Using a CDC25B point mutation that cannot interact with CDK, we show that part of CDC25B activity is independent of its action on the cell cycle.

**\*For correspondence:**
fabienne.pituello@univ-tlse3.fr (FP);
eric.agius@univ-tlse3.fr (EA)

†These authors contributed equally to this work

**Competing interests:** The authors declare that no competing interests exist.

## Introduction

In multicellular organisms, managing the development, homeostasis and regeneration of tissues requires the tight control of self-renewal and differentiation of stem/progenitor cells. This issue is particularly evident in the nervous system, where generating the appropriate number of distinct classes of neurons is essential to constructing functional neuronal circuits.

Steadily increasing data reveal links between the cell cycle and stem cells' choice to proliferate or differentiate (*Soufi and Dalton, 2016*). The G1 phase is usually associated with the initiation of differentiation. Notably, the length of the G1 phase has been shown to play a major role in controlling cell fate decisions in neurogenesis, haematopoiesis (*Lange and Calegari, 2010*) and mammalian embryonic stem cells (*Coronado et al., 2013*; *Sela et al., 2012*), including human embryonic stem cells (hESCs) (*Pauklin and Vallier, 2013*; *Sela et al., 2012*). During cortical neurogenesis, a lengthening of the G1 phase is associated with the transition from neural-stem-like apical progenitors (AP) to fate restricted basal progenitors (BP) (*Arai et al., 2011*). Reducing G1 phase length leads to an increased progenitor pool and inhibition of neuronal differentiation, while lengthening G1 phase promotes the opposite effect (*Calegari et al., 2005*; *Pilaz et al., 2009*). In developing spinal cord, G1 phase duration increases with neurogenesis (*Kicheva et al., 2014*; *Saade et al., 2013*). Interestingly, in hESCs and in neurogenesis it has been shown that the stem/progenitor cell uses Cyclin D, which controls G1 phase progression, to directly regulate the signaling pathways and the transcriptional program controlling cell fate choice (*Bienvenu et al., 2010*; *Lukaszewicz and Anderson, 2011*; *Pauklin et al., 2016*; *Pauklin and Vallier, 2013*). A transient increase of epigenetic modifiers

at developmental genes during G1 has also been reported to create 'a window of opportunity' for cell fate decision in hESCs (*Singh et al., 2015*).

Modification of other cell cycle phases has been correlated with the choice to proliferate or differentiate. Work on hESCs reveals that cell cycle genes involved in DNA replication and G2 phase progression maintain embryonic stem cell identity (*Gonzales et al., 2015*), leading the authors to propose that S and G2/M mechanisms control the inhibition of pluripotency upon differentiation. In the amphibian or fish retina, the conversion of slowly dividing stem cells into fast-cycling transient amplifying progenitors with shorter G1 and G2 phases, propels them to exit the cell cycle and differentiate (*Agathocleous et al., 2007*; *Locker et al., 2006*). A shortening of the S phase correlates with the transition from proliferative to differentiating (neurogenic) divisions in mouse cortical progenitors (*Arai et al., 2011*). In the developing spinal cord, shorter S and G2 phases are associated with the neurogenic phase (*Cayuso and Martí, 2005*; *Kicheva et al., 2014*; *Le Dréau et al., 2014*; *Molina and Pituello, 2017*; *Peco et al., 2012*; *Saade et al., 2017*, *Saade et al., 2013*; *Wilcock et al., 2007*). Until now these links between cell cycle kinetics and cell fate were most often correlations, with the direct impact of cell cycle modifications on cell fate choice being only indirectly addressed. The strong correlations between the cell cycle machinery and the stem cell's choice in different model systems, emphasize the importance of elucidating how these systems work.

Notably, in developing neuroepithelia, cell cycle is synchronized with an oscillatory nuclear movement called Interkinetic Nuclear Migration (INM). Nuclei of progenitor cells occupy specific positions according to cell cycle phase: nuclei migrate basally in the G1 phase, so that the S phase occurs on the basal side, and apically in the G2 phase, allowing mitosis to happen at the apical surface (*Molina and Pituello, 2017*; *Norden et al., 2009*). In mouse corticogenesis and in the chicken neural tube, it was shown that nuclei migrate apically during G2 using the dynein/microtubule motor system (*Baffet et al., 2015*; *Spear and Erickson, 2012*). It has been established that a key cell cycle regulator, the cyclin-dependent kinase 1 (CDK1), triggers dynein recruitment to nuclear pores leading to apical nuclear movement during G2 phase (*Baffet et al., 2015*). The mechanisms involved in basalward migration in G1 are more controversial, ranging from a passive and stochastic process driven by a crowding effect to a movement triggered by microtubule/kinesin3 or actomyosin cytoskeleton (*Miyata et al., 2014*; *Molina and Pituello, 2017*; *Spear and Erickson, 2012*). In the chicken spinal cord, zebrafish retina and rat neocortex, it has been proposed that progenitor nuclei position along the apico-basal axis could lead to a differential exposure to proliferative or differentiative signals that could in turn regulate progenitors cell fate (*Carabalona et al., 2016*; *Del Bene, 2011*; *Del Bene et al., 2008*; *Murciano et al., 2002*).

A link has previously been established between a regulator of the G2/M transition, the CDC25B phosphatase and neurogenesis (*Gruber et al., 2011*; *Peco et al., 2012*; *Ueno et al., 2008*). The cell division cycle 25 family (CDC25) is a family of dual specificity phosphatases that catalyze the dephosphorylation of CDKs, leading to their activation and thereby cell cycle progression (*Aressy and Ducommun, 2008*). Three CDC25s A, B, C have been characterized in mammals, and two, CDC25s A and B have been found in chick (*Agius et al., 2015*; *Boutros et al., 2007*). As observed for numerous cell cycle regulators, these molecules are tightly regulated at the transcriptional and post-transcriptional levels (*Boutros et al., 2007*). The N-terminal region of CDC25B contains the regulatory domain, and the C-terminal region hosts the catalytic domain and the domain of interaction with known substrates, the CDKs (*Sohn et al., 2004*). In Xenopus, CDC25B loss-of-function reduces the expression of neuronal differentiation markers (*Ueno et al., 2008*). An upregulation of CDC25B activity associated with precocious neurogenesis has been observed in an animal model of microcephaly (*Gruber et al., 2011*). Using the developing spinal cord as a paradigm, we previously reported that CDC25B expression correlates remarkably well with areas where neurogenesis occurs (*Agius et al., 2015*; *Peco et al., 2012*). We showed that reducing CDC25B expression in the chicken neural tube alters both cell cycle kinetics, by increasing G2-phase length, and neuron production (*Agius et al., 2015*; *Peco et al., 2012*). However, it is not clear whether the change in cell cycle kinetics is instrumental in cell fate change.

The aim of the present study is to further understand the mechanisms by which CDC25B promotes neurogenesis. First, we use a neural specific loss-of-function in mice to show that Cdc25b is also required for efficient neuron production in mammals. Second, we use gain- and loss-of-function in chicken to show that CDC25B is necessary and sufficient to promote neuron production by controlling the mode of division. We directly measured CDC25B effects upon modes of division, using

recently developed biomarkers that allow differentiating with single-cell resolution proliferative versus neurogenic divisions in the developing spinal cord (*Le Dréau et al., 2014*; *Saade et al., 2017*; *Saade et al., 2013*). We also carried out a clonal analysis using the Brainbow strategy (*Loulier et al., 2014*). Both approaches show that CDC25B decreases proliferative divisions and promotes neurogenic divisions. In addition, we show that CDC25B controls the switch from slow to fast nuclei departure to the basal side during early G1 in the proliferative population. A mathematical model of these dynamics suggests that the cell cycle duration is not instrumental in the observed evolution of the mode of division.

Furthermore, to directly address the putative role of cell cycle kinetics on the mode of division, we use a point mutated form of CDC25B, CDC25B$^{\Delta CDK}$ unable to interact with CyclinB/CDK1 complex. We show that this molecule affects basal G1 movement, neurogenic divisions and neuronal differentiation, even though it does not affect the duration of the G2 phase.

## Results

### Genetic *Cdc25b* invalidation induces a G2-phase lengthening and impedes neuron production in the mouse developing spinal cord

We previously showed that downregulating CDC25B levels using RNAi in the chicken neural tube results in a G2 phase lengthening and a reduction of the number of neurons (*Peco et al., 2012*). Here we used a genetic approach to question whether both functions are conserved in mammals, using a floxed allele of *Cdc25b* and a *NestinCre;Cdc25b*$^{+/-}$ mouse line to specifically ablate the phosphatase in the developing nervous system (*Figure 1A*). In the mouse embryo, *Cdc25b* is detected in the neural tube from E8.5 onward and remains strongly expressed in areas where neurogenesis occurs, as illustrated in the E11.5 neural tube (*Figure 1B*). Loss of *Cdc25b* mRNA was observed from E10.5 onward in *NestinCre;Cdc25b*$^{fl/-}$ embryos (Cdc25b$^{nesKO}$, *Figure 1B*). We therefore determined the consequences of the Cre-mediated deletion of the floxed *Cdc25b* allele on cell cycle parameters and neurogenesis starting at E11.5.

The proliferation capacity of the neural progenitors in *NestinCre;Cdc25b*$^{fl/-}$ embryos, was determined by quantification of EdU labelled replicating neural progenitors. The proliferative index in the dorsal spinal cord (number of EdU+ cells among total number of neural progenitors labelled with Pax7 antibody) was similar between *NestinCre;Cdc25b*$^{fl/-}$ and control embryos (*NestinCre;Cdc25b*$^{fl/+}$ or *Cdc25b*$^{fl/+}$ or *Cdc25b*$^{fl/-}$) (*Figure 1C*). Similarly, the fraction of mitotic cells assessed by quantifying the number of Phospho-Histone 3 (PH3) mitotic cells in the Pax7+ cells displayed a slight and non-significant reduction in the mitotic index of mutant embryos (*Figure 1D*). Since downregulating CDC25B in the chicken neural tube resulted in a lengthening of the G2 phase, we next compared the length of the G2 phase in the dorsal spinal cord of *NestinCre;Cdc25b*$^{fl/-}$ versus control embryos using the percentage of labeled mitosis (PLM) (*Quastler and Sherman, 1959*). Embryos were injected with EdU and allowed to recover for 1 hr, 2 hr or 3 hr before fixation and staining with EdU and PH3 antibodies. We found that the percentage of PH3/EdU positive cells was consistently lower in the dorsal domain of *NestinCre;Cdc25b*$^{fl/-}$ versus control embryos (*Figure 1E*). The average G2-lengths extracted from the curve were 2 hr 19 min in mutants compared to 1 hr 49 min in controls (*Figure 1E*). This indicates that *Cdc25b* loss-of-function in dorsal neural progenitors results in a G2 phase lengthening.

The question is then whether *Cdc25b* loss-of-function affects spinal neurogenesis. Neuron production occurs in two phases in the dorsal spinal cord, an early neurogenic phase (between E9.5 and E11.5) and a late neurogenic phase (between E11.5 and E13.5) (*Hernandez-Miranda et al., 2017*). Neurons emerging from the dorsal spinal cord express numerous transcription factors including Pax2 and Tlx3 that label distinct neuron types and when combined, identify different subtypes of early (Pax2: dl4, dl6; Tlx3: dl3, dl5) and late born neurons (Pax2: dILA; Tlx3: dILB). The use of a *NestinCre* mouse line allows us to accurately ablate the phosphatase at the time of late neuron production (*Hernandez-Miranda et al., 2017*). We hence analyze the impact of the deletion at E11.5 and E12.5. At E11.5, the number of Tlx3+ cells is reduced in the *NestinCre;Cdc25b*$^{fl/-}$ compared to control embryos. Pax2+ neurons are also reduced yet non-significantly (*Figure 1F,G*). One day later, a clear and significant reduction of 25.7% and 28% in the number of Pax2+ and Tlx3+ neurons,

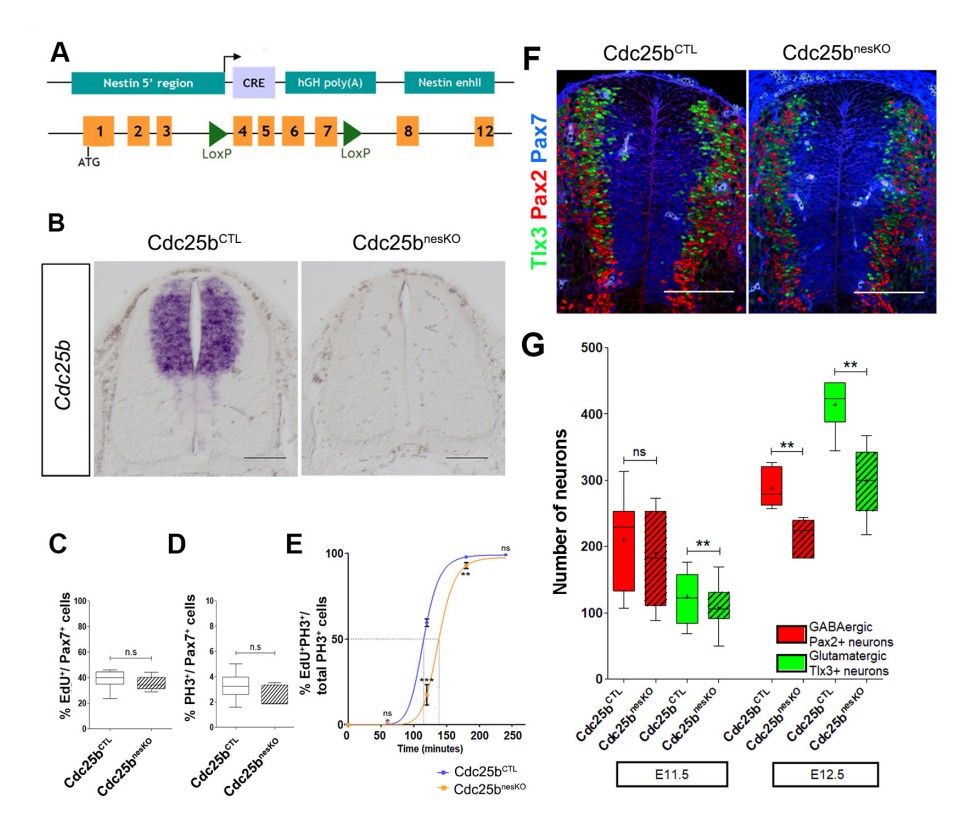

**Figure 1.** *Cdc25b* conditional genetic loss-of-function increases the G2-phase length and impairs dorsal spinal neurogenesis. (A) Scheme of the genetic construction for *Cdc25b* conditional loss-of-function. (B) *Cdc25b* in situ hybridization at E11.5 in control (Cdc25b$^{CTL}$) and conditional nesKO (Cdc25b$^{nesKO}$) conditions. (C–D) Box plots (5/95 percentile) comparing the proliferative index: distribution of the percentage of EdU$^+$/Pax7$^+$ cells indicative of the rate of S-phase cells at E11.5 in control and nesKO neural tubes (C), distribution of the percentage of PH3$^+$/Pax7$^+$ cells indicative of the mitotic index at E11.5 in control and nesKO neural tubes (D). The proliferative index was analyzed using 20 control and seven nesKO embryos. (E) Progression of the percentage of EdU$^+$PH3$^+$/total PH3$^+$ labeled nuclei with increasing EdU exposure time in control and nesKO conditions. The dashed lines correspond to 50% EdU$^+$/PH3$^+$ cells and indicate the G2 length. (F) Cross-sections of E12.5 embryo neural tubes, stained with Pax7, Pax2 and Tlx3 immunostaining in control and nesKO conditions. (G) Box plots (5/95 percentile) comparing the distribution of the number of Pax2 and Tlx3 neurons in control and nesKO conditions at E11.5 and E12.5. The number of analyzed embryos was 15 control vs 11 nesKO for Pax2 and 15 control vs 10 nesKO for Tlx3. The cross indicates the mean value. Mixed model, ** p<0.01. Scale bar represents 100 μm.

The online version of this article includes the following figure supplement(s) for figure 1:

**Figure supplement 1.** Cdc25b conditional genetic loss-of-function affects the progenitor pool.

respectively, is observed following *Cdc25b* deletion. Progenitor domain size measured at E11.5 and E12.5 using Pax7 or Sox2 immunohistochemistry shows non-significant differences (*Figure 1—figure supplement 1*). Analysis of dorsal progenitor nuclear density using DAPI shows a small and constant 7.5% and 7.3% reduction at E11.5 and E12.5, respectively, in mutant versus control embryos (*Figure 1—figure supplement 1*), indicating that neuronal reduction is at least partly due to this small reduction in the progenitor population. Quantification of active caspase three immunostaining (E12.5) does not reveal an increase in cell death, showing that the reduction in neuron number is not due to apoptosis (not shown). The ratio of dILA to dILB neurons is similar in control (0.68) and mutant embryos (0.71), confirming that *Cdc25b* does not impact specific neuronal cell type but rather has a generic effect on neuron production. Together, these observations demonstrate that efficient spinal neuron production requires CDC25B in mammalian embryos and illustrate that this function is conserved among higher vertebrates.

## CDC25B gain-of-function increases neuronal production

The fact that CDC25B downregulation impedes neuron production in mouse and chicken embryos, prompted us to test whether CDC25B gain-of-function is sufficient to stimulate neurogenesis. It is not possible to perform CDC25B gain-of-function using a robust ubiquitous promoter, because an unscheduled increase of the phosphatase during the cell cycle leads to mitotic catastrophe and subsequent apoptosis (*Peco et al., 2012*). To circumvent this technical impasse, we express CDC25B using the mouse cell cycle dependent CDC25B cis regulatory element (ccRE) that reproduces the cell cycle regulated transcription of CDC25B (*Körner et al., 2001*) and prevents apoptosis (*Kieffer et al., 2007*). We verify that ccRE is sufficient to drive lacZ reporter expression in the entire chicken neural tube after transfection by in ovo electroporation (*Figure 2—figure supplement 1A*). Under the control of ccRE, the eGFP-CDC25B fusion protein is expressed in a subset of transfected cells (*Figure 2A*). The level of chimeric protein detected results from the periodic expression induced by the promoter and the intrinsic instability of CDC25B actively degraded at the end of mitosis. The fusion protein can be observed both in the nucleus and cytoplasm of neuroepithelial progenitors located close to the lumen (L) and in mitotic progenitors (*Figure 2A*, arrowhead). The gain-of-function does not induce apoptosis, as revealed by quantification of active caspase three immunostaining (*Figure 2—figure supplement 1B–D*). To ascertain that the phosphatase is functional, we analyze its impact on G2 phase duration. As expected, ectopic expression of the phosphatase shortens the G2 phase (*Figure 2B*, blue curve) without significantly modifying the mitotic index or the proliferation index (*Figure 2—figure supplement 1E–F*).

Quantitative analysis performed on the entire neural tube using NeuroD-reporter assay indicates that increasing CDC25B is sufficient to promote neuronal commitment (*Figure 2—figure supplement 2*). In the neural tube, development of the ventral progenitor population is usually considered more advanced than its dorsal counterpart (*Kicheva et al., 2014*; *Saade et al., 2013*). Accordingly, the temporality of neuron production progresses from ventral to dorsal (*Kicheva et al., 2014*; *Saade et al., 2013*) and correlates with endogenous *CDC25B* expression (*Peco et al., 2012*). We therefore analyze separately the fraction of neurons generated following CDC25B gain-of-function in the ventral and dorsal halves of this structure. In the ventral neural tube, CDC25B gain-of-function increases the percentage of HuC/D$^+$ GFP$^+$ cells from 61.6 ± 1.5% to 76.5 ± 0.9%. Similarly, in the dorsal spinal cord, the proportion increases from 30.7 ± 1.34% to 41.8 ± 2.64% with the CDC25B gain-of-function (*Figure 2F,G*). A significant increase in neurogenesis is also observed using Pax2 immunostaining, from 11.4 ± 1% to 20 ± 1.8% (*Figure 2C,D*). Conversely, CDC25B gain-of-function reduces the proportion of cells expressing the progenitor marker Sox2 (*Figure 2E*). Together, these results indicate that CDC25B is sufficient to stimulate neuron production.

## CDC25B has no effect on mitotic spindle parameters

An increase in CDC25B activity has been shown to induce a shifted cleavage plane and precocious neurogenesis during corticogenesis in mouse (*Gruber et al., 2011*). We therefore tested the effect of CDC25B gain-of-function on spindle orientation in spinal neural precursors. We measured the angle of the mitotic spindle as previously described (*Saadaoui et al., 2014*), and we did not observe a significant change in spindle orientation (*Figure 3A,B*). Another element implicated in asymmetric cell fate in neural progenitors is the spindle-size asymmetry (SSA), that is, the difference in size between the two sides of the spindle (*Delaunay et al., 2014*). Our CDC25B gain-of-function experiments did not induce a significant modification of the SSA in chick spinal neural progenitors (*Figure 3C–D*). In summary, our analyses did not reveal an effect of CDC25B activity on the orientation or the size of the mitotic spindle.

## CDC25B downregulation maintains proliferative divisions and hinders neurogenic divisions

To elucidate CDC25B function, we investigate whether it promotes neurogenesis by controlling the division mode of neural progenitors. We take advantage of a strategy recently developed by E. Marti and colleagues, described to distinguish the three modes of division, PP, PN and NN, occurring in the developing chicken spinal cord (*Le Dréau et al., 2014*; *Saade et al., 2017*; *Saade et al., 2013*). Briefly, the neural tube is electroporated with Sox2::GFP and Tis21::RFP reporters, and 24 hr later the number of neural progenitors expressing these markers is quantified at mitosis. Thus,

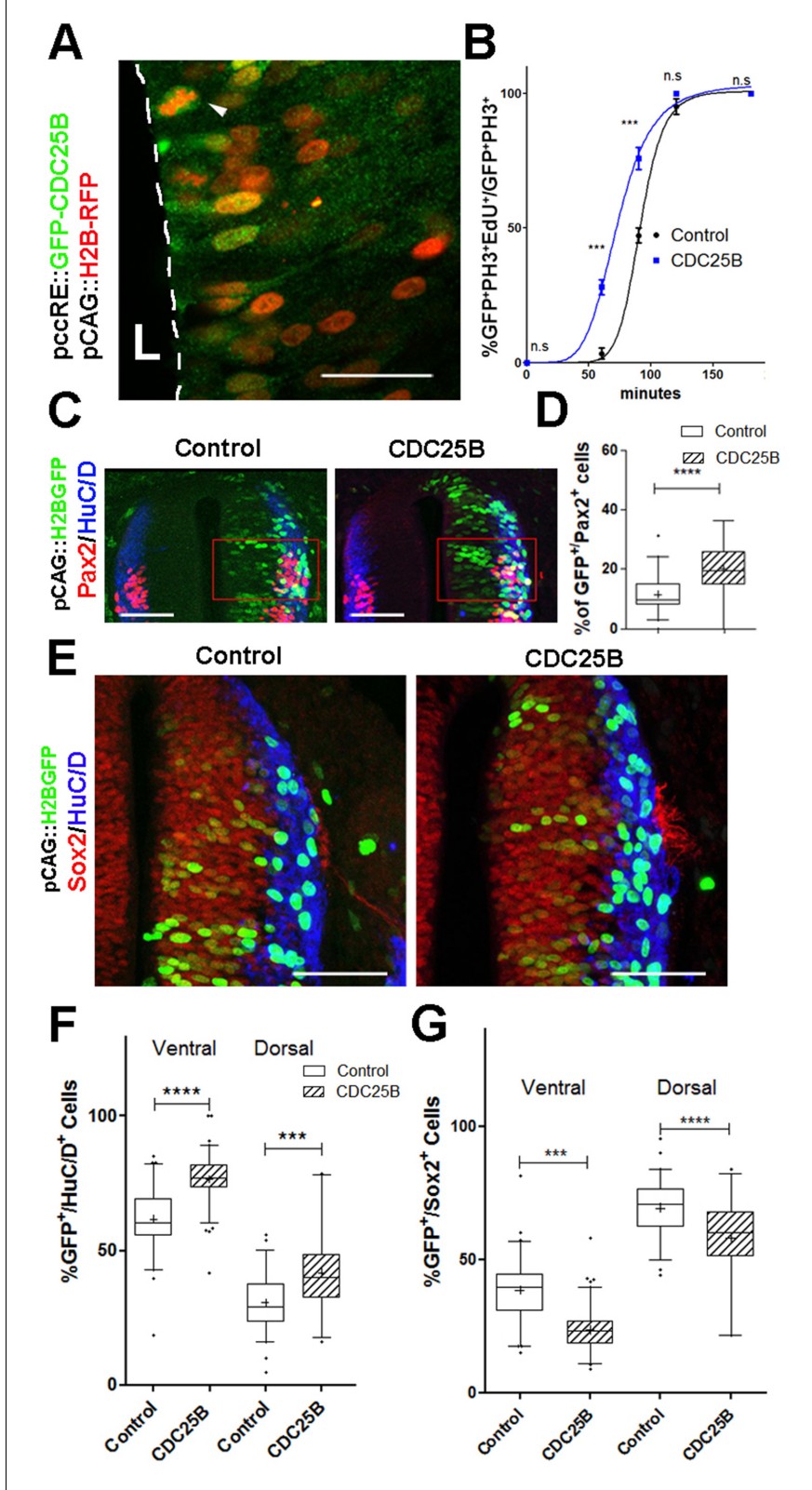

**Figure 2.** CDC25B speeds up neuronal production. (**A**) Cross section of E2.5 chick spinal cord 24 hr after electroporation of pCAG::H2B-RFP vector and pccRE::GFP-CDC25B vector, followed by an anti-GFP immunolocalisation. Note that the protein is expressed in the dorsal neuroepithelium in cells exhibiting a nucleus close to the lumen side (L) or undergoing mitosis (arrowhead). Scale bar indicates 50 μm. (**B**) Curves representing

*Figure 2 continued on next page*

*Figure 2 continued*

the percentage of electroporated GFP$^+$EdU$^+$PH3$^+$ over the total GFP$^+$PH3$^+$ cells with increasing EdU exposure times: control (black), CDC25B gain of function (blue). Note that the curve corresponding to the CDC25B condition (blue) is shifted to the left, showing a reduction in G2 phase length. (C) Representative sections of E3.5 chick spinal cord 48 hr after co-electroporation of a pCAG::H2B-GFP with either a pccRE::lacZ (control) or a pccRE::CDC25B expression vector and processed for Pax2 (red) and HuC/D (blue) immunostaining. The red box illustrates the quantified domain. Scale bars indicate 100 μm. (D) Box plots (5/95 percentile) comparing the percentage of Pax2$^+$ cells within the electroporated population in the control and CDC25B gain-of-function experiments in the dorsal neural tube. Data from three different experiments with eight embryos for the control, and five embryos for the CDC25B gain-of-function. (E) Representative sections of E3.5 chick spinal cord 48 hr after co-electroporation of pCAG::H2B-GFP with either a control or a CDC25B expression vector and processed for Sox2 (red) and HuC/D (blue) immunostaining. Scale bars indicate 100 μm. (F) Box plots (5/95 percentile) comparing the percentage of electroporated HuC/D$^+$ cells in the ventral and dorsal neural tube. Data represent three different experiments with a total of 13 dorsal and six ventral embryos for the control, and 6 dorsal and seven ventral embryos for the CDC25B gain-of-function. The cross represents the mean value. (G) Box plots (5/95 percentile) comparing the percentage of Sox2$^+$ cells within the electroporated dorsal or ventral neural tube in the control, and CDC25B gain-of-function. Same conditions as in F.

The online version of this article includes the following figure supplement(s) for figure 2:

**Figure supplement 1.** CDC25B gain-of-function does not increase apoptosis, S or M cell cycle phase lengths.
**Figure supplement 2.** Effects of various CDC25B constructs on NeuroD promoter activity.

---

Sox2$^+$Tis21$^-$ cells expressing only Sox2::GFP correspond to PP divisions, Sox2$^-$Tis21$^+$ cells correspond to NN divisions, while cells co-expressing both biosensors Sox2$^+$Tis21$^+$ correspond mostly to asymmetric neurogenic divisions, PN (*Figure 4A*). Using these biomarkers in the dorsal neural tube, we obtained comparable results to the ones previously described (Figure 2B in *Le Dréau et al., 2014*). Because the number of electroporated cells in mitosis is very small, we determine whether counting neural progenitors displaying green, yellow or red fluorescence is equivalent to counting only mitotic cells in the dorsal spinal cord 24 hr post electroporation. We do not detect a significant difference in the distribution of cells in total neuroepithelial progenitors (55.4 ± 6.2% Sox2$^+$Tis21$^-$ cells, 29.3 ± 3.9% Sox2$^+$Tis21$^+$ cells and 15.2 ± 2.9% Sox2$^-$Tis21$^+$ cells) and during mitosis (57.9 ± 9.3% Sox2$^+$Tis21$^-$ cells, 23.2 ± 8.5% Sox2$^+$Tis21$^+$ cells and 19 ± 7.3% Sox2$^-$Tis21$^+$ cells) (*Figure 4B*). Because of reporter stability, the temporal window of analysis of Sox2/Tis21 reporters is restricted to 24 hr (*Saade et al., 2013*). Recent data indicate that Sox2 mRNA expression can be sustained in some neurogenic progenitors (*Albert et al., 2017*). A fraction of the Sox2$^+$Tis21$^+$ cells might therefore correspond to NN rather than PN divisions, suggesting that the use of these biomarkers is not sufficient to separate PN and NN divisions in our experimental conditions. Therefore, even if we quantified separately the three populations, we considered the Sox2$^+$Tis21$^+$ and the Sox2$^-$Tis21$^+$ progeny as a whole, producing neurogenic divisions, and compared it to the Sox2$^+$-Tis21$^-$ cells performing proliferative divisions.

*CDC25B* RNAi electroporation leads to a consistent and strong downregulation in *CDC25B* transcripts located in the intermediate neural tube (*Figure 4C*, bracket). We therefore determine the impact of *CDC25B* downregulation on the mode of division in progenitors located in this domain. We co-electroporate the biomarkers with either the *CDC25B*-RNAi plasmid, or the control scrambled plasmid at stage HH11, and quantify the distribution of Sox2$^+$Tis21$^-$, Sox2$^+$Tis21+ and Sox2$^-$Tis21$^+$ cells 24 hr later at stage HH17 (*Figure 4D–E*). When compared to the control scrambled RNAi, the *CDC25B* RNAi induces a massive increase in Sox2$^+$Tis21$^-$ progeny (13.4 ± 1.3% to 35.1 ± 1.8%), and a decrease in Sox2$^+$Tis21$^+$ progeny (from 72.1 ± 1.85% to 56.2 ± 1.70% and to some extent, in Sox2$^-$Tis21$^+$ progeny (from 14.6 ± 1.43% to 8.74 ± 0.8%, *Figure 4E*). Therefore *CDC25B* RNAi increases proliferative divisions from 13.4 ± 1.3% to 35.1 ± 1.8% and decreases neurogenic divisions from 86.6 ± 1.3% to 63.9 ± 1.8% (P-value < 0,0001). This observation indicates that CDC25B downregulation hinders neuron production by maintaining proliferative divisions and reducing neurogenic divisions.

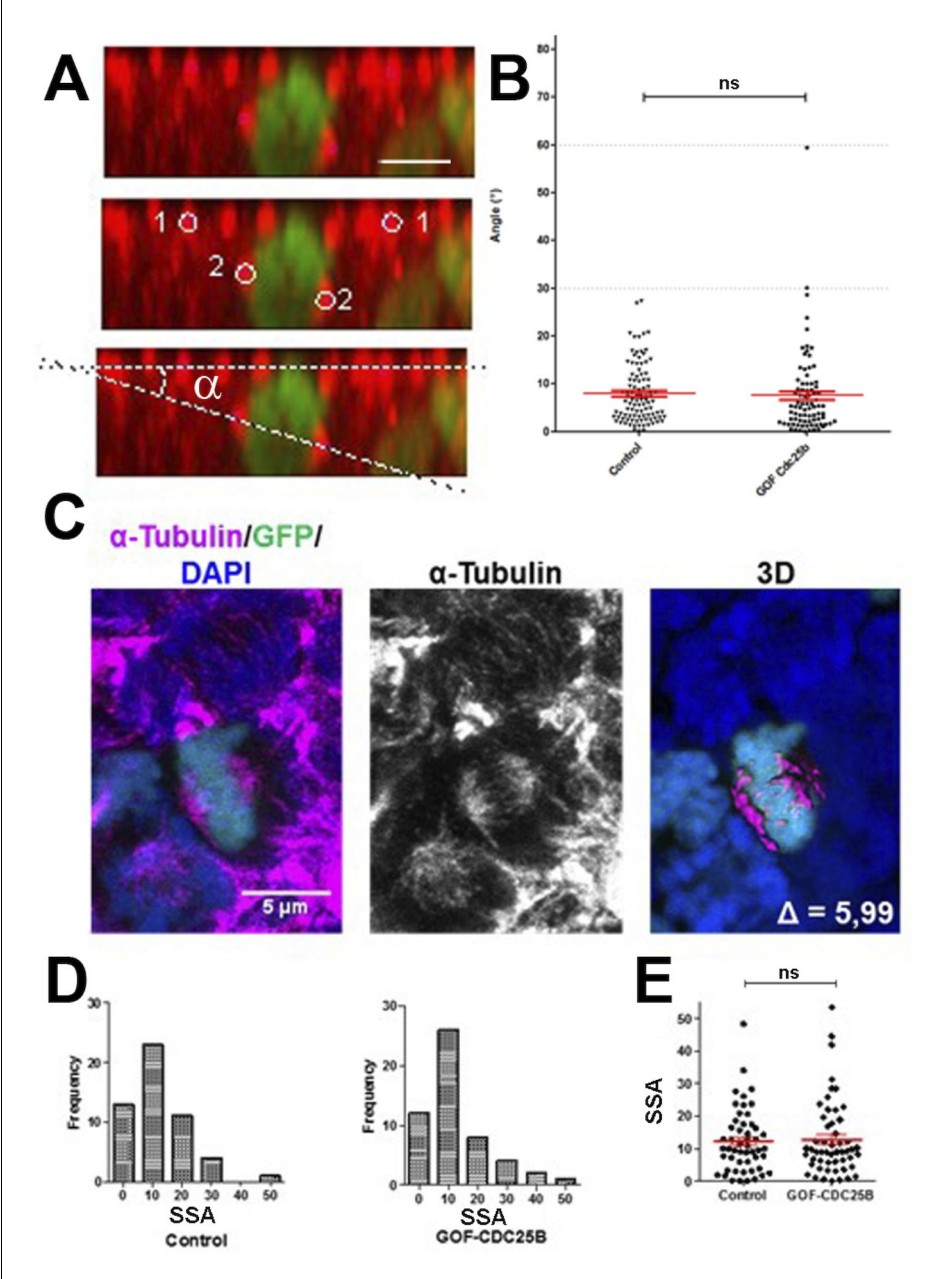

**Figure 3.** CDC25B gain-of-function does not affect mitotic spindle orientation or spindle-size asymmetry (SSA). (**A**) Representative Z plane image of an anaphase cell expressing H2B-GFP that decorates chromosomes (green) and immunostained with anti γ tubulin antibody to label centrosomes (red). Aligned interphase centrosomes labelled as one and mitotic spindle poles labelled as 2 (middle image) were used to measure mitotic spindle angle $\alpha$ (lower image). (**B**) Quantification of mitotic spindle angle $\alpha$, 24 hr after electroporation in control and CDC25B gain-of-function experiments. (**C**) Representative image of a symmetric metaphase cell: H2B-GFP and DAPI stain the nuclei, and $\alpha$-Tubulin stains the mitotic spindle (left and middle images). Right image, 3D reconstruction of the symmetric spindle using Imaris software. (**D, E**) Distribution of the Spindle-Size Asymmetry (SSA) difference between the two sides of the spindle 24 hr after electroporation in control and CDC25B gain-of-function: Histogram of SSA distribution (**D**) and scatter plot of SSA distribution (**E**). Scale bars represent 5 μm.

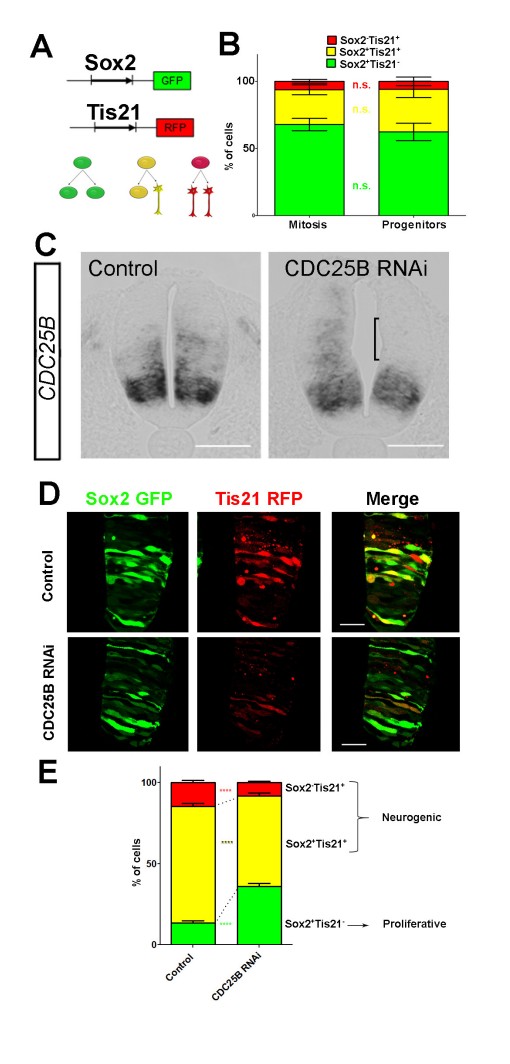

**Figure 4.** CDC25B downregulation reduces neurogenic divisions. (**A**) Schematic representation of the Sox2:: GFP Tis21::RFP labelling strategy. A GFP expressing cell (green cell) corresponds to a PP division, a cell expressing both GFP and RFP (yellow cell) corresponds to a PN division, and a RFP expressing cell (red cell) corresponds to a NN division. (**B**) Bar plot representing the percentage of cells expressing the reporters Sox2:: GFP and Tis21::RFP at HH17 in the entire progenitor population, or in progenitors performing mitosis identified with phospho-histone-3 (PH3) immunostaining. Note that these results are not significantly different. These data are obtained from three different experiments, seven embryos, 365 progenitors, and 79 mitoses. (**C**) In situ hybridization for *CDC25B* on HH17 spinal cord, 24 hr post electroporation of Control RNAi (left panel) and CDC25B RNAi (right panel). The reduction of CDC25B expression in the intermediate region is indicated by a bracket. Cells were electroporated on the right side of the neural tube (not shown). Scale bars indicate 100 μm. (**D**) Cross-sections of chick spinal cord at HH17, 24

*Figure 4 continued on next page*

## CDC25B Gain-of-function promotes neurogenic divisions

We then tested how CDC25B gain-of-function affects the mode of division using two different approaches: the Tis21/Sox2 assay (*Saade et al., 2013*) (*Figure 5A,D,F*), and a clonal analysis using the Brainbow technique (*Tozer et al., 2017*) (*Figure 5B,C,E*). Analyzing the distribution of the mode of division with the Tis21/Sox2 strategy in the entire neuroepithelium at 24 hr after electroporation, showed that CDC25B leads to a decrease in Sox2$^+$Tis21$^-$ progeny (from 46.1 ± 2.1% to 27.3 ± 1.8%), an increase in Sox2$^+$-Tis21$^+$ progeny (from 43.4 ± 1.9% to 50.9 ± 1.6%), and Sox2$^-$Tis21$^+$ progeny (from 10.5 ± 0.8% to 21.8 ± 1.5%, *Figure 5A,D*). This shows that CDC25B gain-of-function reduces proliferative divisions from 46.1 ± 2.1% to 27.3 ± 1.8% and increases neurogenic divisions from 53.8 ± 2.1% to 72.7 ± 1.8% (P-value < 0,0001). Embryos co-transfected with the Nucbow vector (*Loulier et al., 2014*), limiting amounts of Cre recombinase (*Morin et al., 2007*) and the various gain-of-function constructions, were harvested after 40 hr at stage HH21 and labelled using HuC/D immunostaining (*Figure 5B,C*). Two cells clones located in low electroporated density area were selected on the basis of color identity (*Tozer et al., 2017*) and categorized as proliferative divisions: HuC/D$^-$-HuC/D$^-$ (upper panel in *Figure 5C*), and neurogenic divisions either HuC/D$^+$-HuC/D$^-$(middle panel in *Figure 5C*) or HuC/D$^+$-HuC/D$^+$ (lower panel in *Figure 5C*).

Using this alternative strategy, we showed that the expression of CDC25B leads to a decrease in HuC/D$^-$-HuC/D$^-$ cells (from 76.5 ± 2.6% to 60.7 ± 2.3%) and to an increase in HuC/D$^-$-HuC/D$^+$ cells (from 9.4 ± 1.5% to 16.3 ± 1.7%) and HuC/D$^+$-HuC/D$^+$ (from 14.1 ± 2.3% to 23.0 ± 2.1%, *Figure 5E*). We observe a decrease in proliferative divisions from 76.5 ± 2.6% to 60.7 ± 2.3% and an increase of neurogenic divisions from 23.8 ± 2.5% to 39.4 ± 2.3% (P-value < 0.001). While the mode of division repartitions were probably different due to inherent differences in the two strategies, CDC25B gain-of-function results in consistent modifications in both assays : a reduction of 15.8 percentage-points (pp) (from 76.5% to 60.6%) and 18.8 pp (from 46.1% to 27.3%) in proliferative divisions and a corollary increase of 15.8 pp (from 23.7% to 36.4%) and 18.8 pp (from 53.8% to 72.7%) in neurogenic divisions in the Nucbow and the Tis21/Sox2 assays, respectively (*Figure 5D,E*). These results indicate that CDC25B gain-of-function in

hr after co-electroporation of Sox2::GFP and Tis21::RFP reporter, plus a control RNAi vector or the CDC25B-RNAi vector. Scale bars indicate 50 µm. (E) Bar plot representing the percentage of progenitors expressing Sox2::GFP and Tis21::RFP 24 hr after co-electroporation of a control vector or a CDC25B RNAi vector. 4 experiments include seven control embryos and 15 CDC25B RNAi embryos.

spinal neural progenitors reduces proliferative divisions and promotes neurogenic divisions.

As previously described, neurogenesis progresses from ventral to dorsal in the developing spinal cord. Accordingly, at the electroporation time (stage HH11), the neural tube contains essentially self-expanding progenitors (*Le Dréau et al., 2014*; *Saade et al., 2013*). 24 hr later, (stage HH17), the repartition of the modes of division is not the same in dorsal and ventral control conditions. Dorsal neural

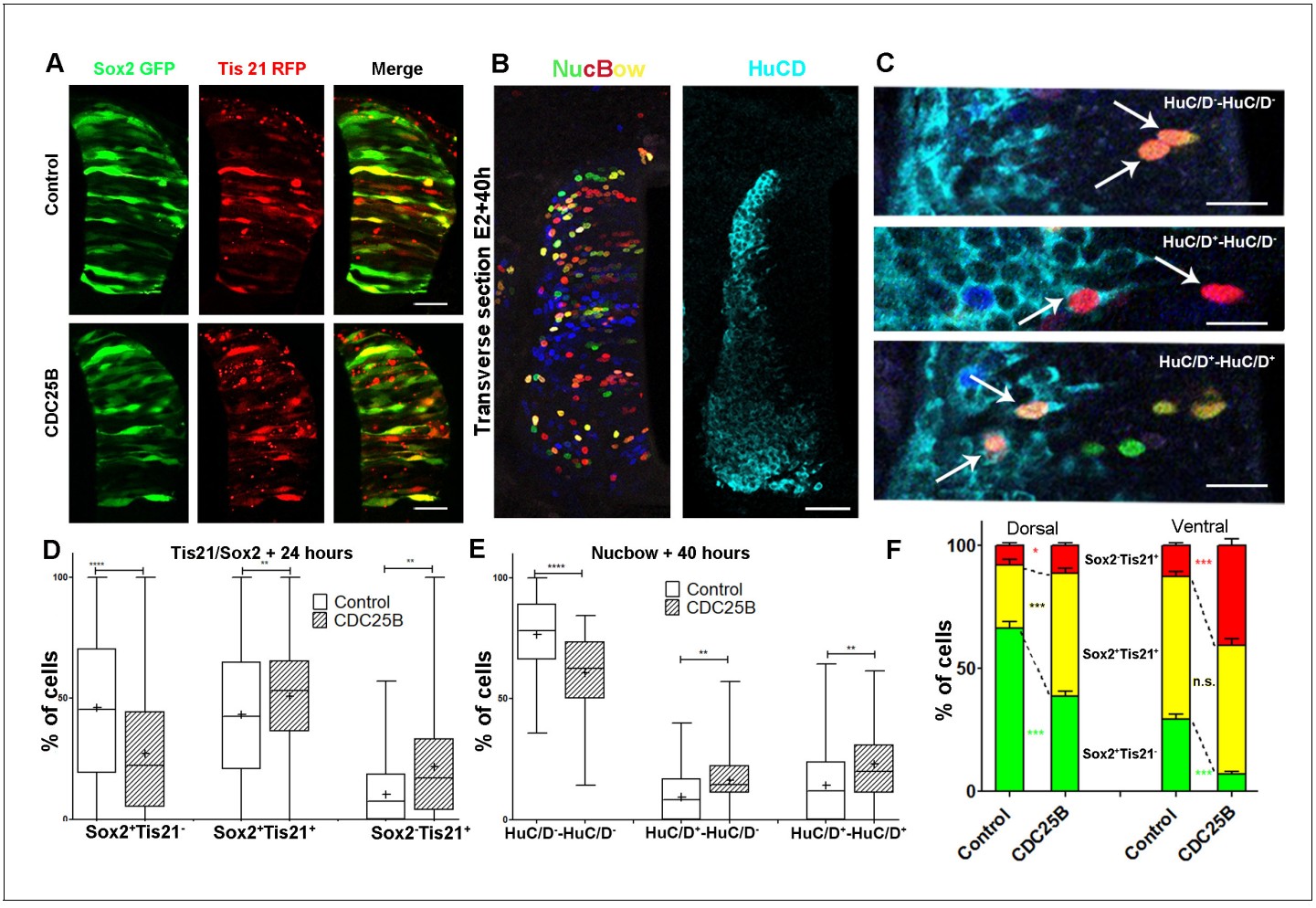

**Figure 5.** CDC25B gain-of-function promotes neurogenic divisions. (A) Representative cross-sections of HH17 chick spinal cord, 24 hr after electroporating Sox2::GFP and Tis21::RFP reporters, plus a control vector pccRE::lacZ, or a pccRE::CDC25B vector. Scale bars indicate 50 µm. (B) Representative cross-sections of HH21 chick spinal cord, 40 hr after electroporation of Nucbow and pCX CRE vectors, and immunostaining with HuC/D antibody. Scale bar indicates 50 µm. (C) Specific two cell clone examples, 40 hr after transfection of Nucbow and immunostaining with HuC/D antibody. Scale bars indicate 10 µm. (D) Box plots (5/95 percentile) comparing the percentage of progenitors expressing Sox2::GFP and Tis21::RFP 24 hr after co-electroporation with control or CDC25B vectors in the entire spinal cord. Data represent the means ± SEM of 3 different experiments with 5 control and 6 CDC25B gain-of-function embryos. (E) Box plots (5/95 percentile) comparing the percentage of two cell clones expressing Nucbow and pCX CRE vectors, 40 hr after co-electroporation with control or CDC25B vectors in the entire spinal cord. Data represent the means ± SEM of 3 different experiments with 387 clones in 12 control embryos, and 659 clones in 11 CDC25B gain-of-function embryos. (F) Bar plot representing the percentage of progenitors expressing Sox2::GFP and Tis21::RFP 24 hr after co-electroporation with control or CDC25B vectors in the dorsal and ventral spinal cord. Data represent the means ± SEM. Data represent three different experiments with 5 dorsal and 10 ventral neural tubes in the control, and 5 dorsal and six ventral neural tubes in the CDC25B gain-of-function.

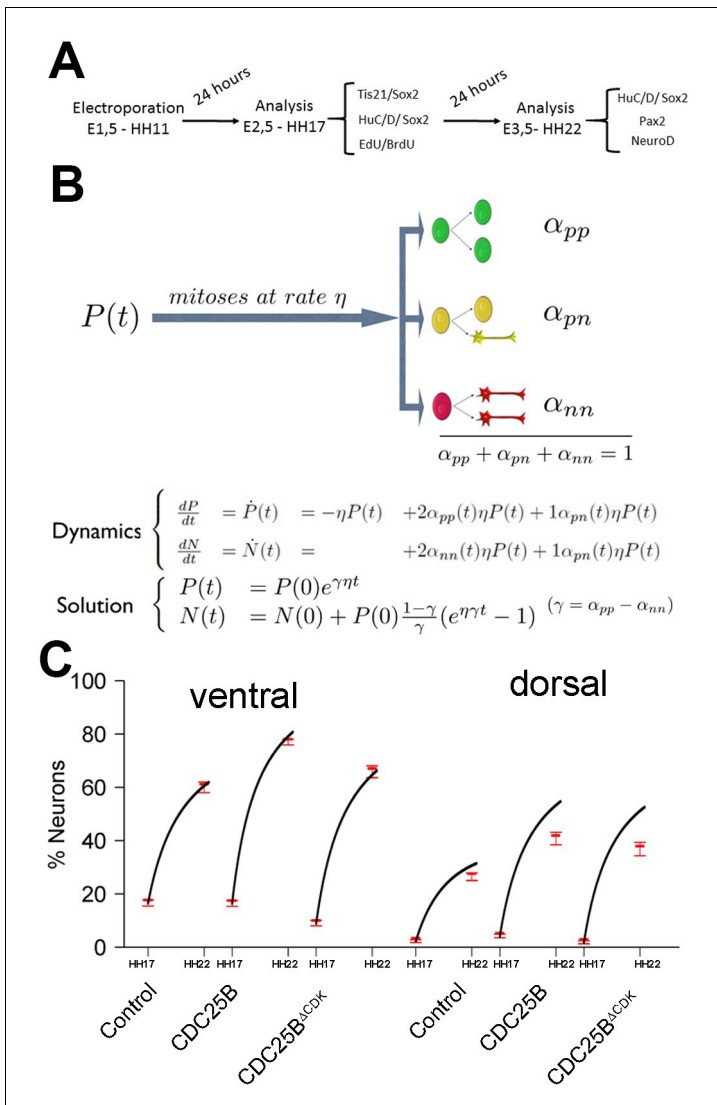

**Figure 6.** Mathematical model linking the mode of division to the fraction of neurons generated. (**A**) Scheme of the experimental time course. Neural tubes are electroporated at stage HH11. 24 hr (HH17) and 48 hr (HH22) post electroporation, cell cycle parameters, mode of division and progenitor/neuronal markers are analyzed. (**B**) Illustration of our mathematical model. We consider P(t) a pool of progenitors at a given time with a mitotic rate $\eta$. These mitoses lead to three modes of division: a fraction $\alpha_{pp}$ producing symmetric proliferative divisions yielding two progenitors, a fraction $\alpha_{pn}$ producing asymmetric divisions yielding one progenitor and one neuron (a precursor of), and a fraction $\alpha_{nn}$ producing symmetric neurogenic divisions yielding two neurons. The equations display the dynamics governing the pools of progenitors P(t) and neurons N(t) at any time t. These dynamics are solved for a given initial condition P(0), N(0), and we obtain the state of the system any time later (Solution, details in Appendix 2 and Appendix 4). (**C**) Kinetic predictions of the neuronal fraction between stages HH17 and HH22 in the different conditions, compared to the mean ±95% confidence interval (in red) of the experimental data at stages HH17 and HH22 (from Figures *Figure 2F* and *Figure 7E*).

tube contains mainly self-expanding progenitors (66.3% Sox2⁺Tis21⁻ cells, *Figure 5A,F*) (*Le Dréau et al., 2014*), whereas ventral neural tube encloses essentially neurogenic progeny (58% of Sox2⁺-Tis21⁺ cells and 12.7% of Sox2⁻Tis21⁺ cells, *Figure 5A,F*) (*Saade et al., 2013*). Because of this difference, we analyze the effects of CDC25B on the dorsal and ventral neural tube separately. In the dorsal neural tube, CDC25B gain-of-function leads to a reduction in the percentage of Sox2⁺Tis21⁻ progeny (from 66.3 ± 2.6% to 38.6 ± 2.1%) and a concomitant increase in the percentage of Sox2⁺-Tis21⁺ progeny (from 25.9 ± 2.1% to 50.1 ± 1.9%). In this tissue, the percentage of Sox2⁻Tis21⁺

progeny progresses only slightly (from 7.8 ± 1.2 to 11.3 ± 1%, *Figure 5F*). This observation indicates that CDC25B gain-of-function in early steps of neurogenesis reduces proliferative Sox2$^+$Tis21$^-$ progeny and increases Sox2$^+$Tis21$^+$ neurogenic progeny. In the ventral neural tube, CDC25B gain-of-function induces a massive reduction in Sox2$^+$Tis21$^-$ progeny (from 29.3 ±1.3% to 6.9 ± 1%) and leads to an increase in Sox2$^-$Tis21$^+$ progeny (from 12.7 ±1.1% to 40.7 ± 2.7%), without significantly modifying the percentage of Sox2$^+$Tis21$^+$ progeny (from 58 ± 2% to 52.3 ± 2.8%, *Figure 5F*). Thus, CDC25B ectopic expression in a more advanced neural tissue reduces proliferative divisions and increases Sox2$^-$Tis21$^+$ neurogenic progeny.

Together, these results suggest that CDC25B activity in neural progenitors reduces proliferative divisions and promotes neurogenic divisions, depending on the receiving neural tissue.

## Mathematical modelling reveals that cell cycle duration is not instrumental in controlling the mode of division

To test quantitatively data from a dynamical point of view (*Míguez, 2015*; *Saade et al., 2013* ; Appendix Neurogenic decisions require a cell cycle independent function of the CDC25B phosphatase), we formalized in mathematical terms our current understanding of what happens in this biological system (*Figure 6A*). Despite the fact that a fraction of the Sox2$^+$Tis21$^+$ cells might correspond to NN rather than PN divisions, in the modeling part below, we assumed that the Sox2/Tis21 reporter expression is indicative of PP, PN and NN as described in (*Saade et al., 2013*).

We consider a population of progenitors at time $t_0$, $P(t_0)$, and we assumed that their different modes of division result in expanding either the pool of progenitors $P(t)$ through proliferative divisions (PP divisions), or the pool of neurons $N(t)$ by neurogenic divisions (PN and NN divisions). Denoting $\eta$ the rate at which P cells undergo divisions per unit time (which depends only on the cell cycle duration), the growth rates of the two pools only depend on the relative magnitude of each mode of division.

Denoting $\alpha_{pp}$, $\alpha_{pn}$ and $\alpha_{nn}$ the corresponding proportions of the modes of division (their sum is 1), the growth rates of the two pools (i.e. their time derivatives $\dot{P}(t)$ and $\dot{N}(t)$ for Progenitors and Neurons respectively) can then be directly formalized as:

$$\begin{cases} \dot{P}(t) & = -\eta P(t) & +2\alpha_{pp}\eta P(t) + 1\alpha_{pn}\eta P(t) \\ \dot{N}(t) & = & +2\alpha_{nn}\eta P(t) + 1\alpha_{pn}\eta P(t) \end{cases} \tag{1}$$

In this model, the evolution of the pool of progenitors is governed by $\alpha_{pp}$ and $\alpha_{nn}$ (because $\alpha_{pn}$ does not affect the pool of progenitors, only the pool of neurons). Denoting $\gamma = \alpha_{pp} - \alpha_{nn}$ the difference between the two proportions, we then have $\gamma = 1$ when $\alpha_{pp} = 1$, $\alpha_{nn} = 0$, corresponding to purely self-expanding progenitors and $\gamma = -1$ when $\alpha_{pp} = 0$, $\alpha_{nn} = 1$, corresponding to fully self-consuming progenitors. Hence $\gamma$ is a good indicator of the balance between proliferation and differentiation of the progenitors (*Míguez, 2015*). Using $\gamma$, the model can be rewritten more simply as:

$$\begin{cases} \dot{P}(t) & = \gamma\eta P(t) \\ \dot{N}(t) & = (1-\gamma)\eta P(t) \end{cases}$$

An explicit solution, for $\gamma \neq 0$, is:

$$\begin{cases} P(t) & = P(0)e^{\gamma\eta t} \\ N(t) & = N(0) + P(0)\frac{1-\gamma}{\gamma}(e^{\eta\gamma t} - 1) \end{cases} \tag{3}$$

This equation means that if the quantities of progenitors and neurons are determined at a given time ($P(0)$, $N(0)$), for example at HH17, we can compute the expected number of progenitors and neurons at any time later, for example at HH22, provided that the modes of division and cell cycle times can be considered constant over the considered period. Full details of the mathematical work and statistics are given in Appendix 2. We then compare quantitatively the experimental data to the predictions based on our current hypotheses. This comparison is surprisingly auspicious for the control and gain-of-function experiments in the ventral zone (*Figure 6C*, left). In this zone, considering the ratio between the two pools at HH17 (e.g. the measured fractions of neurons), the measured cell cycle duration (12 hr), the set of modes of division measured at HH17, and the hypothesis that those

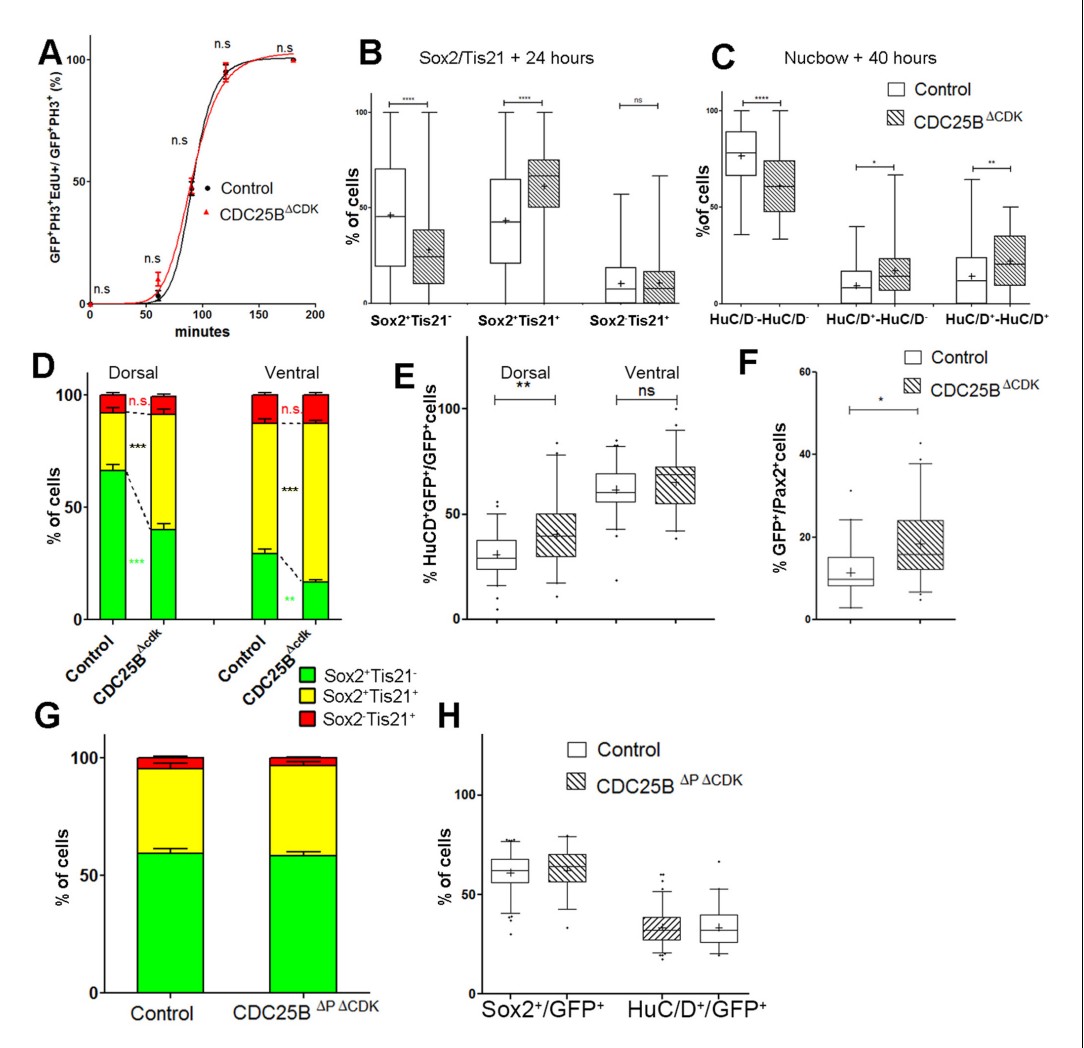

**Figure 7.** CDC25B gain-of-function promotes neurogenesis independently of CDK interaction. (**A**) Curves representing the percentage of electroporated GFP$^+$EdU$^+$PH3$^+$ over the total GFP$^+$PH3$^+$ cells with increasing EdU exposure times: control (black), CDC25B$^{\Delta CDK}$ (red). Note that the curve for the CDC25B$^{\Delta CDK}$ condition is similar to the control, indicating an absence of effect on G2 length. (**B**) Box plots (5/95 percentile) comparing the percentage of progenitors expressing Sox2::GFP and Tis21::RFP 24 hr after co-electroporation with control or CDC25B$^{\Delta CDK}$ vectors in the entire spinal cord. Data represent the means ± SEM of 3 different experiments with 6 control and 6 CDC25B$^{\Delta CDK}$ gain-of-function embryos. (**C**) Box plots (5/95 percentile) comparing the percentage of two cell clones expressing Nucbow and pCX CRE vectors, 40 hr after co-electroporation with control or CDC25B$^{\Delta CDK}$ vectors in the entire spinal cord. Data represent the means ± SEM of 3 different experiments with 387 clones in 12 control embryos, and 692 clones in 10 CDC25B$^{\Delta CDK}$ gain-of-function embryos. (**D**) Bar plot representing the percentage of cells expressing Sox2::GFP and Tis21::RFP 24 hr after co-electroporation with control or CDC25B$^{\Delta CDK}$ vectors, in the dorsal or ventral spinal cord. Data represent the means ± SEM. Data represent three different experiments with a total of 5 dorsal and 10 ventral neural tubes under control conditions, and 4 dorsal and 9 ventral neural tubes in CDC25B$^{\Delta CDK}$ gain-of-function. (**E**) Box plots (5/95 percentile) comparing the percentage of HuC/D$^+$ cells within the electroporated population in control or CDC25B$^{\Delta CDK}$ gain-of-function experiments, in the dorsal or ventral neural tube at HH22. Data represent three different experiments with 13 dorsal and 6 ventral neural tubes in the control and 6 dorsal and 3 ventral neural tubes in the CDC25B$^{\Delta CDK}$ gain-of-function. (**F**) Box plots (5/95 percentile) comparing the percentage of Pax2$^+$ cells in the dorsal neural tube at HH22. Data from three different experiments with 8 control embryos, and 11 CDC25B$^{\Delta CDK}$ gain-of-function embryos. (**G**) Bar plot representing the percentage of progenitors expressing Sox2::GFP and Tis21::RFP at HH17, 24 hr after electroporation of a control or CDC25B$^{\Delta P \Delta CDK}$ expressing vector in the dorsal half of the spinal cord. Data from three different experiments with 6 control embryos, and 9 CDC25B$^{\Delta P \Delta CDK}$ embryos. (**H**) Box plots (5/95

*Figure 7 continued on next page*

*Figure 7 continued*

percentile) comparing the percentage of Sox2$^+$ or HuC/D$^+$ cells within the electroporated population in the control or CDC25B$^{\Delta P\Delta CDK}$ gain-of-function experiments, in the dorsal spinal cord at HH22. Data from three different experiments with 11 control embryos, and 6 CDC25B$^{\Delta P\Delta CDK}$ embryos. The cross indicates the mean value.

modes of divisions stay unmodified during 24 hr, the model predicts with good accuracy the ratios between the two pools at HH22. In the dorsal zone, the model correctly predicts the control condition, and it confirms the tendency of CDC25B gain-of-function to promote a greater neuron fraction, albeit with some quantitative discrepancy (the model overestimates the fraction of neurons). This suggests that, notwithstanding biological complexity, the general picture of a pool of progenitors among which cells undergo stochastic modes of division, appears relevant.

Our model is built on the assumption that all cells undergo asynchronous mitosis at the same rate, and that the fate of any mitosis is stochastic and probabilistically distributed according to the fraction of dividing cells undergoing PP, PN or NN divisions, namely a common division rate for all progenitors associated with probabilistic fates (Appendix 3). In this picture, the proportion of mode of division controls directly the numbers of progenitors and neurons that are generated. However, the model is compatible with an extreme alternative interpretation, in which the three modes of division correspond to specific division rates associated with deterministic fates (Appendix 3). In this case, each population of progenitors has a specific mean cycling time and the cell cycle time is instrumental to the mode of division. Namely, cycling at rate $\alpha_{pp}\eta$ would result in a PP division, cycling at rate $\alpha_{pn}\eta$ would result in a PN division, and cycling at rate $\alpha_{nn}\eta$ would result in a NN division. Therefore, the numbers and proportions of progenitors/neurons at HH22 would result from the difference between cell cycle times associated with modes of division. We compute these putative cell cycle times based on the data obtained in the three conditions and the two zones (*Table 1*). The wide range of specific cycle times, that is, from 17 to 172.7 hr, is incompatible with data usually recorded (reviewed in *Molina and Pituello, 2017*). This suggests that, in the time window of our analyses, the observed evolution of progenitors and neurons cannot be exhaustively explained by pure differences in cell cycle durations among the three modes of division.

## CDC25B promotes neurogenesis independently of CDK interaction

One prediction of our model is that neurogenesis might be affected independently of cell cycle length modification. To test whether the CDC25B-induced G2 phase modification is instrumental in promoting neurogenesis, we use a mutated form of CDC25B that was shown not to affect cell cycle kinetics. The mutation prevents CDC25B-CDK1 interactions without affecting CDC25B phosphatase activity (*Sohn et al., 2004*). Accordingly, expressing this mutated form of the phosphatase called CDC25B$^{\Delta CDK}$, does not modify G2 phase length in neuroepithelial progenitors (*Figure 7A*, red curve). The effects of CDC25B$^{\Delta CDK}$ on the division mode are then compared in the entire neuroepithelium using the Tis21/Sox2 approach, 24 hr after electroporation, and the Nucbow technique, 40 hr after electroporation.

In the Tis21/Sox2 strategy, CDC25B$^{\Delta CDK}$ decreases Sox2$^+$Tis21$^-$ progeny (from 46.1 ± 2.1% to 28 ± 1.6%) and increases Sox2$^+$Tis21$^+$ progeny (from 43.4 ± 1.9% to 61.2 ± 1.5%). The percentage of Sox2$^-$Tis21$^+$ progeny is not modified in that experimental context (from 10.5 ± 0.8% to 10.8 ± 0.8%, *Figure 7B*). These data show that CDC25B$^{\Delta CDK}$ gain of function reduces proliferative divisions from 46.1 ± 2.1% to 28 ± 1.6% and increases neurogenic divisions from 53.8 ± 2.1% to 72 ± 1.6% (P-value < 0,0001). Analyzes using the Nucbow strategy, show a decrease in HuC/D$^-$-HuC/D$^-$ cells (from 76.5 ± 2.6% to 61 ± 2.4%), an increase in HuC/D$^-$-HuC/D$^+$ cells (from 9.4 ± 1.5% to 17.1 ± 2.5%) and in HuC/D$^+$-HuC/D$^+$ cells (from 14.1 ± 2.3% to 21.9 ± 2.1%, *Figure 7C*) following CDC25B$^{\Delta CDK}$ expression. Therefore, CDC25B$^{\Delta CDK}$ gain of function reduces proliferative divisions from 76.5 ± 2.6% to 61 ± 2.4% and increases neurogenic divisions from 23.8 ± 2.5% to 39 ± 2.4% (P value < 0,001). CDC25B$^{\Delta CDK}$ gain-of-function results in an equivalent reduction of 15.8 and 18.1 percentage-point for proliferative and a corollary increase in neurogenic divisions in the Nucbow and Sox2/Tis21 strategies respectively (*Figure 7B,C*).

**Table 1.** Putative time it would take to achieve the three kinds of division under a model which assumes that only cycle time determines the fate output.

Full consequences derived from this assumption are given in Appendix 3. Basically, such an assumption implies that cycling rates associated with each mode of division are proportional to the observed fraction of that mode. If we observe, for instance, 60% PP-divisions and 10% NN-divisions (like it is in the Control dorsal), then a NN-division should take six times as long as a PP-division. If we exclude such a possibility, then fate distribution cannot be exclusively determined by differences in fate-based cycle times. It does not exclude that a given kind of fate (e.g. proliferative divisions PP) could require a longer time to be achieved than others; it excludes that such differences would suffice per se to explain the differences between the fractions of fates.

| Zone and condition | $T_{pp}$ (hours) | $T_{pn}$ (hours) | $T_{nn}$ (hours) | $T_c$ (hours) |
|---|---|---|---|---|
| Control dorsal neural tube | 18.1 | 46.3 | 154.1 | 12.0 |
| CDC25B dorsal neural tube | 31.1 | 23.9 | 106.0 | 12.0 |
| CDC25B$^{\Delta CDK}$ dorsal neural tube | 29.8 | 23.5 | 150.0 | 12.1 |
| Control ventral neural tube | 41.0 | 20.7 | 94.5 | 12.0 |
| CDC25B ventral neural tube | 172.7 | 22.9 | 29.5 | 12.0 |
| CDC25B$^{\Delta CDK}$ ventral neural tube | 72.2 | 17.0 | 94.7 | 12.0 |

Together these data show that a mutated form of CDC25B unable to interact with CDKs, still promotes neurogenic divisions.

We then analyze the effects of CDC25B$^{\Delta CDK}$ on the dorsal and ventral progenitors 24 hr after electroporation. CDC25B$^{\Delta CDK}$ gain-of-function in the dorsal neural tube, reduces Sox2$^+$ Tis21$^-$ progeny (from 66.3 ± 2.7% to 40.2 ± 2.5%), increases Sox2$^+$Tis21$^+$ progeny (from 25.9 ± 2.1% to 51.1 ± 2.2%), and has no effect in Sox2$^-$Tis21$^+$ progeny (from 7.8 ± 1.2% to 8.0 ± 1.1%, **Figure 7D**). In this context, the fraction of HuC/D$^+$ neurons generated 40 hr following CDC25B$^{\Delta CDK}$ expression increases from 30.7 ± 1.3% to 40.4 ± 2.5%. (**Figure 7E**). Similarly, the percentage of Pax2+ neurons is increased from 11.3 ± 1% to 18.3 ± 1.3% (**Figure 7F**). In the ventral neural tube, CDC25B$^{\Delta CDK}$ overexpression leads to a reduction in Sox2$^+$Tis21$^-$ progeny (29.3 ± 2.1% vs 16.6 ± 1.2%), an increase in Sox2$^+$Tis21$^+$ progeny (58 ± 2% vs 70.7 ± 1.4%) and no effect on Sox2$^-$Tis21$^+$ progeny (12.7 ± 1.1% vs 12.7 ± 1.1%, **Figure 7D**). In the ventral neural tube, CDC25B$^{\Delta CDK}$ induces a slight but non-significant increase in HuC/D expression (**Figure 7E**). We examined our mathematical model to determine whether this slight increase in neuron production is coherent with the fact that the mutated form does not promote Sox2$^-$Tis21$^+$ progeny, and the number of neurons predicted is in agreement with the experimental data (**Figure 6C**).

To determine whether CDC25B$^{\Delta CDK}$ function on neurogenic divisions and neuronal differentiation requires phosphatase activity, we use a form of the protein containing an additional point mutation inactivating the catalytic domain (CDC25B$^{\Delta P \, \Delta CDK}$). This construct does not affect the mode of division at 24 hr (**Figure 7G**). 48 hr post electroporation this mutated form does not modify NeuroD reporter expression (**Figure 2—figure supplement 2**), the percentage of HuC/D + neurons or the percentage of Sox2+ progenitor's populations (**Figure 7H**), indicating that the phosphatase activity is required for the neurogenic function of CDC25B.

Altogether, these results show that CDC25B$^{\Delta CDK}$ stimulates neurogenic divisions and neuronal differentiation without affecting the duration of the G2 phase. This opens the possibility that the phosphatase possesses a function in addition to its canonical role in cell cycle regulation.

## CDC25B promotes fast nuclei apical departure in early G1 independently of CDK interaction

To go further in our understanding of this cell cycle independent role of CDC25B, we set up a high resolution time-lapse imaging technique that allows real-time tracking of the behaviour of single neural progenitor nuclei during G2/M/G1 phases. To perform live imaging, E2 embryo neural tubes are electroporated with a GFP-tagged version of PCNA (**Leonhardt et al., 2000**), then slice cultures of neural tube explants are performed 6 hr after electroporation and analyzed in live experiments

starting 12 hr later. Using this approach, nuclear movements are tracked in time and space during G2, M and G1 phase (*Figure 8*). As previously described (*Spear and Erickson, 2012*), we observe that mitosis initiates away from the apical side and gets completed against the lumen (*Figure 8A–B*). Interestingly, nuclei in G1 display two types of behaviours: either a newly formed nucleus remains close to the lumen (Ap) or it rapidly migrates away from the apical side towards the basal side (Bs), giving rise to three mitotic patterns Ap/Ap, Ap/Bs, Bs/Bs (*Figure 8A–C* and *Video 1*, *Video 2*, *Video 3*). Based on the position of the nuclei 20 min after mitosis (Bs being defined as more than 10 μm away from the apical side at that time), the occurrences of their behaviour were quantified in control and gain-of-function conditions (*Figure 8D*). Under control conditions, Ap represents 50.6 ± 6.9% of the post mitotic behavior, and Bs 49.4 ± 6,9%. CDC25B and CDC25B$^{\Delta CDK}$ gain-of-function decrease Ap (to 24.1 ± 5.3% and 30.9 ± 4.5%, respectively) and increase Bs (to 75.8 ± 5.3% and 69.1 ± 4.5%, respectively).

It is possible that the Bs migratory behavior precedes the apical process withdrawal associated with the onset of neuronal differentiation (*Das and Storey, 2014*; *Tozer et al., 2017*). We took advantage of our time-lapse set up to identify cells re-entering S phase, by the appearance of a dot-like staining within the nuclei corresponding to the recruitment of PCNA into the DNA replication foci (*Figure 8—figure supplement 1*, *Leonhardt et al., 2000*). For all the nuclei whose cell cycle status was identified (*Table 2*), we quantify 9/16, 32/37 and 40/40 Bs nuclei re-entering S phase in Control, CDC25B and CDC25B$^{\Delta CDK}$ gain-of-function, respectively. A majority of Ap nuclei also re-enter S phase in control (30/36), CDC25B (15/21) and CDC25B$^{\Delta CDK}$ (16/20) gain-of-function experiments (*Table 2*). Thus, a majority of Bs and Ap nuclei re-enter S phase in gain-of-function experiments suggesting that the change in migratory behavior is not the consequence of a neurogenic division and of neuronal commitment, but either is upstream or is independent of it.

To analyze more profoundly how CDC25B activity affects nuclei migration in G1, we determined the nuclei motion using a statistical measure of the average distance a nucleus travels over time: the mean squared displacement, MSD (*Norden et al., 2009*). We calculated MSD profiles for the Ap and Bs nuclei under control, CDC25B and CDC25B$^{\Delta CDK}$ gain-of-function conditions (*Figure 8E*). In both cases, the profile clearly exhibits 2 types of motion: Ap nuclei displaying slow motion (diffusion) and Bs nuclei exhibiting directed movements (advection). In addition, we determined the average speed of the nuclei over the first 20 min after mitosis (*Figure 8F*). We observed speeds of 0.26 ± 0.03 μm/min (n = 16), 0.27 ± 0,03 μm/min (n = 17) and 0.27 ± 0,04 μm/min (n = 11) for the Ap nuclei and 1.10 ± 0.17 μm/min (n = 14), 0.98 ± 0.1 μm/min (n = 25) of and 0.80 ± 0.1 μm/min (n = 19) for Bs nuclei in control, CDC25B and CDC25B$^{\Delta CDK}$, respectively.

Together these data suggest that the non-cell cycle dependent activity of CDC25B does not modify much the departure speed of the nuclei, but rather controls the switch from slow to fast nuclei departures from the apical surface.

Altogether, these results show that in the neuroepithelium, the CDC25B phosphatase affects early G1 nuclear behavior, and also is necessary and sufficient to promote neurogenic divisions and neurogenesis. Importantly, CDC25B$^{\Delta CDK}$ without affecting the duration of the G2 phase, still affects early G1 nuclear behavior and stimulates neurogenic divisions and neuronal differentiation. Our results open then the possibility that the phosphatase possesses cell cycle independent and neurogenic functions.

## Discussion

An important issue in the field of neurogenesis concerns the implication of cell cycle function during neuron production (*Agius et al., 2015*). Here, we confirm in mammals our previous observations in birds, that the G2/M cell cycle regulator CDC25B phosphatase is required to finely tune neuronal production in the neural tube. Gain-of-function experiments performed in the chick neural tube reveal that CDC25B activity is sufficient to modify the mode of division of neural progenitors and to promote neuronal differentiation concomitantly with a shortening of the G2 phase length. We demonstrate that CDC25B expression in neural progenitors induces a shift from proliferative to neurogenic divisions and promotes neuronal differentiation independently of any CDK interaction, indicating that it involves a new substrate of the phosphatase (*Figure 9*). Finally, analyses in real time of INM reveal that wild type CDC25B and mutated CDC25B$^{\Delta CDK}$ proteins increase the number

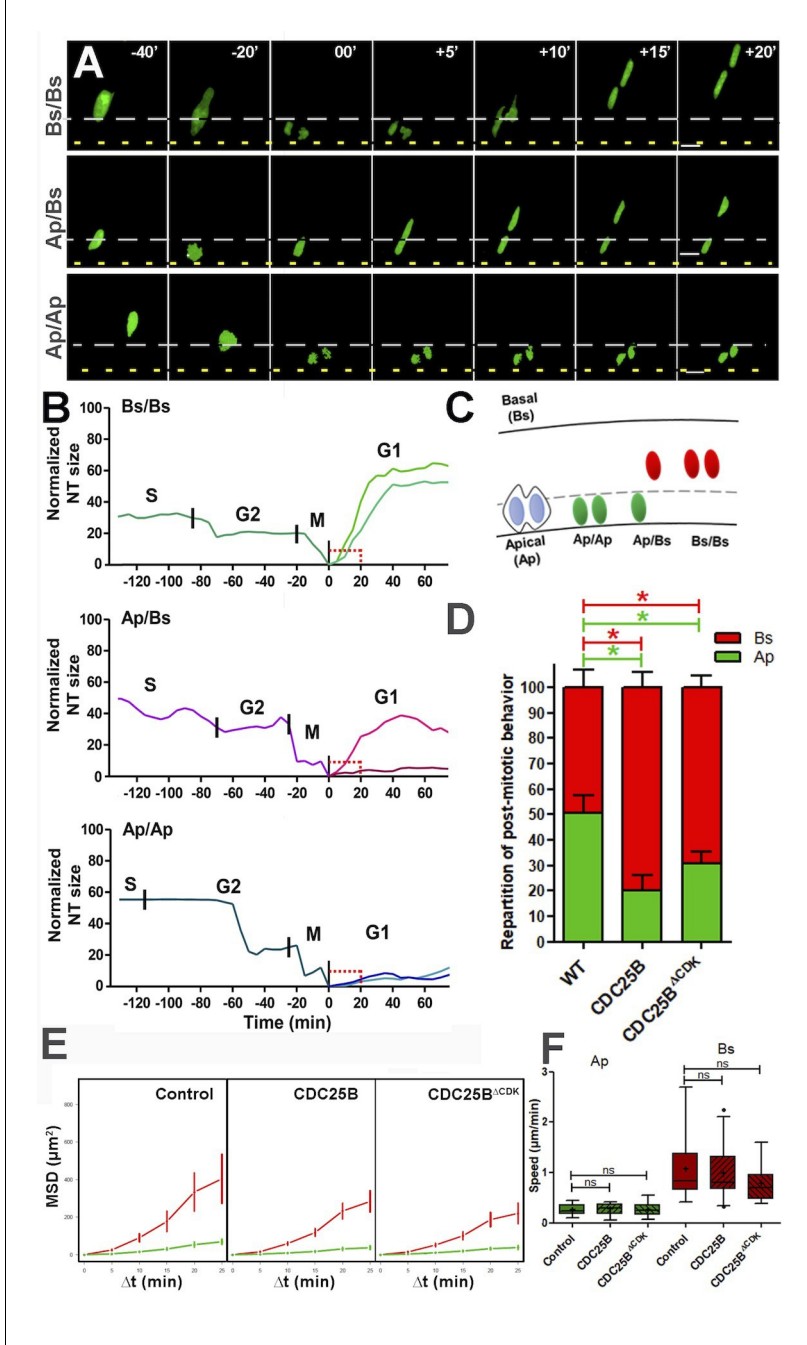

**Figure 8.** CDC25B affects G1 nuclei movement independently of CDK interaction. (**A**) Time-lapse series of the different daughter cell nuclear behaviors. Yellow dashed lines indicate the lumen; grey dashed lines represent 10% of the apico-basal length. Scale bars: 10 µm. (**B**) Quantitative tracking of nuclear movement in embryonic chicken neural tube. Daughter cell nuclei can display three different behaviors after cytokinesis: both nuclei migrate immediately after mitosis (Bs/Bs) (upper panel); one of the nuclei remains at the apical side and the sister nucleus migrates to the basal side (Ap/Bs) (middle panel) or both nuclei remain at the apical side for at least 20 min before starting basal migration (Ap/Ap) (lower panel). Nuclei were labelled by NLS-EGFP-L2-PCNA (*Leonhardt et al., 2000*) that allows the distinction between G2/M/G1 phases, and their movements were tracked by time-lapse microscopy and Imaris software. The end of mitosis (cytokinesis) showed the most apical localization and was defined as zero. Cell cycle phases (S, G2, M, G1) are indicated above the tracks. (**C**) Scheme representing nuclear behavior during G1. (**D**) Quantification of the repartition of post mitotic behavior, that is, Ap or Bs positioning after mitosis in WT, CDC25B and CDC25B$^{\Delta CDK}$ gain-of-function. 156, 174 and 212 cells in 16, 9 and 20 explants of 10, 5 and 8 experiments in WT, CDC25B and CDC25B$^{\Delta CDK}$ gain-of-function, respectively. (**E**) Mean squared

*Figure 8 continued on next page*

*Figure 8 continued*

displacement (MSD) profile (error bars show 95% confidence interval) of Ap nuclei (green line) and Bs nuclei (red line) in the control, CDC25B and CDC25B$^{\Delta CDK}$ gain-of-function. Under all conditions, Ap nuclei display slow motion (linear trend), while Bs nuclei display a persistent apico-basal motion (parabolic trend). (F) Box plots (5/95 percentile) comparing the mean speed over the first 20 min post mitosis of Ap and Bs nuclei. Number of nuclei tracked are 16, 17, and 11 Ap nuclei, and 14, 25, and 19 Bs nuclei, in control and CDC25B and CDC25B$^{\Delta CDK}$ gain-of-function, respectively for E and F.

The online version of this article includes the following figure supplement(s) for figure 8:

**Figure supplement 1.** Time-lapse series of neural progenitor cell electroporated with GFP-PCNA.

of nuclei performing fast basalward movement in early G1, giving us a track to follow in order to elucidate the non-cell cycle function of CDC25B.

## CDC25B is required for efficient neuron production in vertebrates

In mammals three CDC25s (A, B, C) have been characterized, whereas only two CDC25s (A and B) have been found in chicken (*Agius et al., 2015*). In mouse, *Cdc25a* loss-of-function is embryonic lethal, whereas loss-of-function of *Cdc25b* or *Cdc25c* or both has no apparent phenotype except female sterility (*Boutros et al., 2007*). Crossing our floxed mice to ubiquitous Cre:PGK-Cre$^m$(*Lallemand et al., 1998*) also results in female sterility (data not shown). *Cdc25a* has been described as playing a major role in the G1-S transition and is capable of compensating the loss-of-function of the other *Cdc25* members. In the mouse embryonic neural tube, both *Cdc25a* and *Cdc25c* display a broad expression pattern, while *Cdc25b* is mainly expressed in domains where neurogenesis occurs (*Agius et al., 2015* and *Figure 1*). The conditional loss-of-function in the mouse CNS, shows for the first time that *Cdc25b* is involved simultaneously in the control of G2 phase length and spinal neurogenesis. In the mouse, at least part of the reduction in the number of neurons is probably due to the slight reduction in progenitor population. This observation substantiates our data showing that *CDC25B* downregulation, performed using RNAi in chicken embryo, induces a reduction in neurogenesis (*Peco et al., 2012*). Two other studies link *CDC25B* and neurogenesis. First in *Xenopus*, *FoxM1* and *CDC25B* loss-of-function has been shown to reduce expression of neuronal differentiation markers, but not early neuroectoderm markers (*Ueno et al., 2008*). In this context, epistasic analysis shows that *FoxM1* loss-of-function can be rescued by *CDC25B* gain-of-function (*Ueno et al., 2008*). Second, *MCPH1* knock out mice display a microcephalic phenotype due to an alteration of the Chk1-Cdc25-Cdk1 pathway. Indeed, *MCPH1* mutants display a decreased level of the inhibitory Chk1 kinase localized to centrosomes, leading to increased *Cdc25b* and *Cdk1* activities. A premature activation of *Cdk1* leads to an asynchrony between mitotic entry and centrosome cycle. This disturbs mitotic spindle alignment, promoting oblique orientation and precocious neurogenic asymmetric divisions

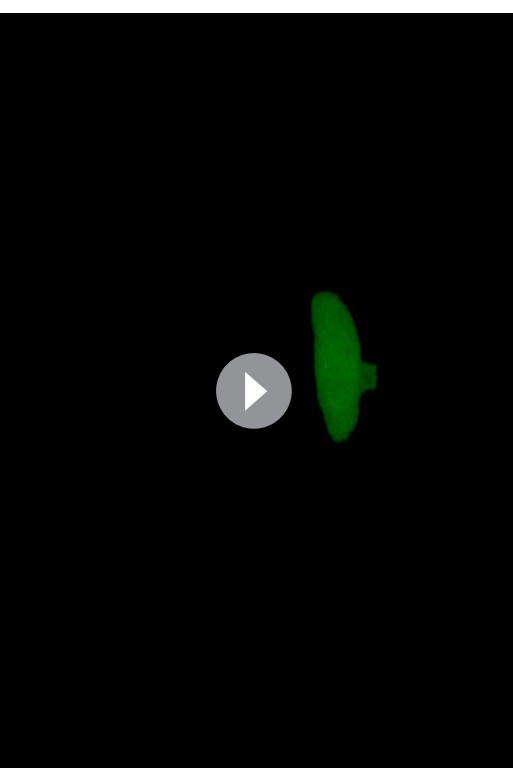

**Video 1.** Time-lapse imaging of neural tube daughter nuclei performing apical movements (Ap/Ap). Mother and daughter cells expressing GFP-PCNA can be followed over time and interkinetic nuclear movement of cells is observed. Images were taken every 5 min at 63X magnification and are played at 12 frames per second (fps).
https://elifesciences.org/articles/32937#video1

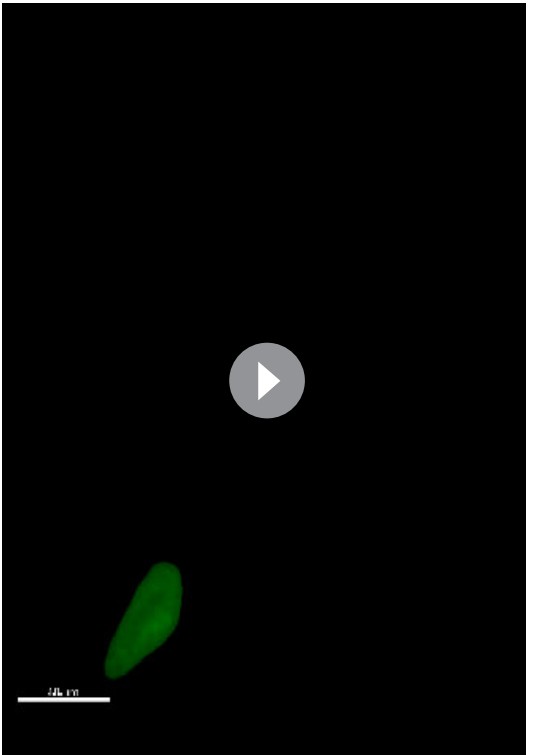

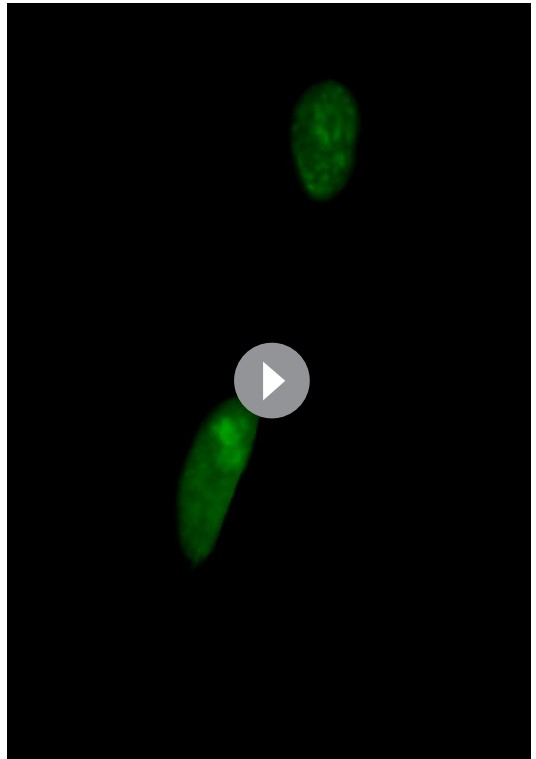

**Video 2.** Time-lapse imaging of neural tube daughter nuclei performing apical and basal movements (Ap/Bs). Mother and daughter cells expressing GFP-PCNA can be followed over time and interkinetic nuclear movement of cells is observed. Images were taken every 5 min at 63X magnification and are played at 12 fps.
https://elifesciences.org/articles/32937#video2

**Video 3.** Time-lapse imaging of neural tube daughter nuclei performing basal movements (Bs/Bs). Mother and daughter cells expressing GFP-PCNA can be followed over time and interkinetic nuclear movement of cells is observed. Images were taken every 5 min at 63X magnification and are played at 12 fps.
https://elifesciences.org/articles/32937#video3

**Table 2.** Distribution of post-mitotic Basal and Apical nuclei performing a new cell division (S-phase) or remaining in G1-phase (Long G1).

n: number of cells counted. S-Phase: cell that re-enters S phase during the time lapse. Long G1: cell that performs a G1 longer that 10 h hours and that does not re-enter S phase during the time lapse. ND: not determined because the time lapse conditions did not allow to follow the cell long enough.

| | S-phase | | Long G1 | | ND | | Total |
|---|---|---|---|---|---|---|---|
| | n | % | n | % | n | % | n |
| | Basal | | | | | | |
| WT | 9 | 11.8 | 7 | 9.2 | 60 | 79.0 | 76 |
| CDC25B | 32 | 25.6 | 5 | 4.0 | 88 | 70.4 | 125 |
| CDC25B$^{\Delta CDK}$ | 40 | 27.2 | 0 | 0.0 | 107 | 72.8 | 147 |
| | Apical | | | | | | |
| WT | 30 | 37.5 | 6 | 7.5 | 44 | 55.0 | 80 |
| CDC25B | 15 | 30.6 | 6 | 12.3 | 28 | 57.1 | 49 |
| CDC25B$^{\Delta CDK}$ | 16 | 24.6 | 4 | 6.2 | 45 | 69.2 | 65 |

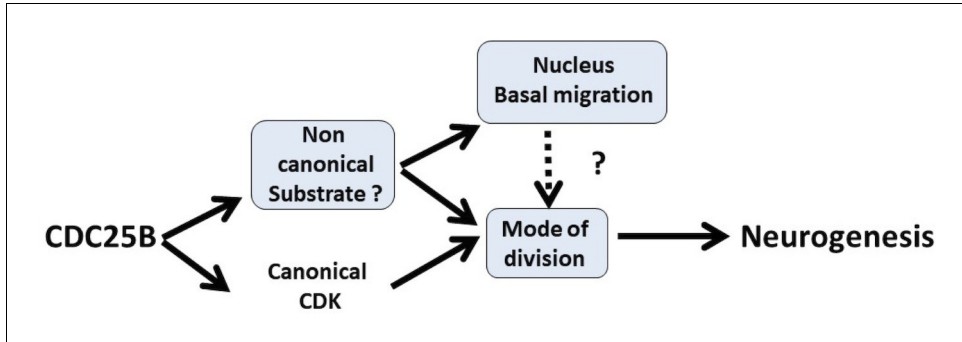

**Figure 9.** Schematic of CDC25B modes of action. CDC25B activity on an unknown substrate changes G1 nucleus basalward movement during Interkinetic Nuclear Migration (INM), and also acts on the mode of division leading to increased neurogenesis. It remains to be determined whether a link exists between these two activities. In addition to this new pathway, the data obtained in mice and using the Tis21/Sox2 assay suggest that the activity of CDC25B on CDK might account for part of its activity on the mode of division and neurogenesis.

(*Gruber et al., 2011*). Moreover, the reduced neurogenic production in the *MCPH1* loss-of-function can be restored by a concomitant Cdc25B loss-of-function, demonstrating the phosphatase's pivotal role in the neurogenic phenotype. Altogether, these observations indicate that *Cdc25b* activity is broadly used during nervous system development among vertebrate species.

## CDC25B promotes neurogenic divisions independently of CDK interaction

CDC25B downregulation reduces the transition from proliferative to neurogenic divisions. To be able to clarify the role of CDC25B on both types of division, we use the cell cycle cis-regulatory element combined with the rapid degradation of CDC25B at the end of M phase, to reproduce the endogenous cyclic expression of the phosphatase (*Körner et al., 2001*). Using Sox2/Tis21 biomarkers and Nucbow clonal analyses, we observe differences in the repartition of the mode of division, probably due to intrinsic methodological differences. This discrepancy may be linked to the differences in the method used: one possibility being that one analysis is performed at 24 hr after electroporation, while the other is performed 40 hr after. Nevertheless, CDC25B gain-of-function reduces proliferative and promotes neurogenic divisions independently of the method used. Gain-of-function of the CDC25B mutated form inactive during the cell cycle, also diminishes proliferative divisions. Using the Sox2/Tis21 biomarkers, we observe different results depending on the population of progenitors targeted.

In the dorsal neural tube, CDC25B gain-of-function increases Sox2$^+$Tis21$^+$ compared to control conditions. In the ventral neural tube, gain-of-function leads to an increase in Sox2$^-$Tis21$^+$, the percentage of Sox2$^+$Tis21$^+$ progeny being unchanged. We propose that ectopic expression of the phosphatase can be interpreted in different ways depending on the context: CDC25B would have the capacity to convert Sox2$^+$Tis21$^-$ into Sox2$^+$Tis21$^+$ in a young tissue, while in an older tissue CDC25B could convert Sox2$^+$Tis21$^-$ into either Sox2$^+$Tis21$^+$ or Sox2$^-$Tis21$^+$. With respect to what occurs in an older tissue, either the phosphatase converts Sox2$^+$Tis21$^-$ into Sox2$^+$Tis21$^+$ or Sox2$^-$Tis21$^+$, or the phosphatase initially promotes Sox2$^+$Tis21$^-$ into Sox2$^+$Tis21$^+$ and subsequently, using the principle of communicating vessels in an older tissue, promotes Sox2$^+$Tis21$^+$ into Sox2$^-$Tis21$^+$ progeny.

We speculate that CDC25B acts as a maturing factor in the progression from stem pool to differentiated neurons, and we suggest that this element of the cell cycle machinery has been coopted to regulate independently cell cycle progression and neurogenesis.

## Mathematical modelling of the neuronal fraction in the dorsal neural tube

The model predicts the ratio of neuron at 48 hr after electroporation, given the ratio at 24 hr and the distributions among the modes of division and the mean cycle length. As a minimal model with

no free parameters, its predictions are still quite well consistent with data in the ventral neural tube for the three conditions. In the dorsal neural tube, while its prediction is also well consistent with data for control condition, it predicts however larger fractions of neurons than those experimentally observed in CDC25B and CDC25B$^{\Delta CDK}$ gain-of-functions (*Figure 6C*), pointing to a missing hypothesis to explain this discrepancy.

We submit several hypotheses. First of all, at HH11, endogenous CDC25B is expressed in the ventral neural tube but not in the dorsal neural tube. This means that electroporation causes a true gain-of-function in the dorsal domain, while in the ventral domain it makes only a dosage modification of a component already present. Then, CDC25B regulation is complex, and an active degradation mechanism in the dorsal neural tube could attenuate the gain-of-function. Another possibility is that electroporated gain-of-function, which is also cell cycle dependent, could be less efficient with time and thereby lead to fewer neurons than expected. Alternatively, the signaling pathway downstream of CDC25B could be expressed differently in the ventral and dorsal neural tubes, and this could limit the gain-of-function effect in the dorsal neural tube. All things considered, we regard the discrepancy between our predictions and our data as a challenging milestone that deserves further investigation. We could have formalized an 'ad hoc' model for each hypothesis mentioned above in order to fit the observed fractions of neurons, yet this would have amounted to add free parameters, and free parameters can always be adjusted at will. We prefer to stress that the standard model for these dynamics still requires identifying further elements in order to reconcile the predictions with the data of this study.

## CDC25B promotes basalward nuclear movement independently of CDK interaction

Here we show for the first time in the spinal cord that nuclei basalward movements occurring in early G1 display two types of motion: slow or fast departure from the apical surface. During mammalian corticogenesis, two not mutually exclusive mechanisms were described for the basal migration of G1 nuclei. It was proposed to be a passive event, depending on a crowding effect due to the apically G2 phase nuclei migration (*Kosodo et al., 2011*). Other studies proposed that the actomyosin system (*Schenk et al., 2009*) or plus-end-directed kinesin/microtubule driven movement (*Tsai et al., 2010*) are involved in carrying nuclei from apical to basal side. In the spinal cord, we show that CDC25B and CDC25B$^{\Delta CDK}$ control the choice between slow or rapid apical departure and promotes the latter. This establishes a new link between a core cell cycle regulator and INM. Similarly a relationship has been described between CDK1 and the minus-end-directed motor dynein in G2 apical movement (*Baffet et al., 2016*; *Hu et al., 2013*). CDK1 phosphorylates the nucleoporin RanBP2, promoting nuclear envelop dynein recruitment.

We observed that the two types of motion in early G1 occur in proliferating progenitors, suggesting that they are either upstream or independent of neurogenesis. A great deal of evidence shows that nuclear movement alterations correlate with neurogenesis modifications, due to alterations in the duration and level of exposure of nuclei to proliferative or differentiation signaling. In zebrafish retina, when the motor protein Dynactin-1 is disrupted, nuclei migrate more rapidly and further into the basal side and more slowly to the apical side. In this context, since Notch signaling is activated on the apical side, mutant progenitors are less exposed to Notch and exit the cell cycle prematurely (*Del Bene et al., 2008*). In the developing rat brain, INM is driven basally by the microtubule motor protein KIF1A, and downregulating KIF1A results in the maintenance of nuclei on the apical side and a severe reduction in neurogenic divisions (*Carabalona et al., 2016*). Radial glial progenitors nevertheless display normal cell cycle progression, indicating that the two events can be uncoupled. The authors propose that this change in nuclear movement increases exposure of neural progenitors to proliferative signals at the apical side, or alternatively keeps the cells further away from differentiating signals. Accordingly, the rapid basal movement induced by CDC25B in the neural tube would reduce exposure of the nucleus to proliferative signals, or expose them to differentiating signals. Interestingly, mouse CDC25A, B and C triple KO (TKO) exhibits epithelial cells in the small intestine blocked in G1 or G2, accompanied by enhanced Wnt signalling activity (*Lee et al., 2009*).

A follow-up to this work could be identifying the CDK independent players downstream of CDC25B. Other CDC25B substrates have been characterised, such as steroid receptors (*Ma et al., 2001*), and the peri-centriolar material component Kizuna (*Thomas et al., 2014*). A recent analysis

using microarrayed Tyr(P) peptides representing confirmed and theoretical phosphorylation motifs from the cellular proteome, identifies more than 130 potential CDC25B substrates (*Zhao et al., 2015*). These substrates are implicated in microtubule dynamics, signalling pathways like Delta/ Notch or Wnt, transcription, epigenetic modifications, mitotic spindle or proteasome activity (*Zhao et al., 2015*), and several of them could play a role in INM or cell fate choice (*Akhtar et al., 2009*; *Aubert et al., 2002*; *Das and Storey, 2012*; *Götz and Huttner, 2005*; *Hämmerle and Tejedor, 2007*; *Jiang and Hsieh, 2014*; *Kimura et al., 2014*; *Li et al., 2012*; *MuhChyi et al., 2013*; *Olivera-Martinez et al., 2014*; *Sato et al., 2004*; *Schwartz and Pirrotta, 2007*; *Vilas-Boas et al., 2011*). Further work will be necessary to dissect the molecular pathway linking CDC25B with INM and to determine whether this link is causal in neurogenesis. In conclusion, we propose that our data illustrate that cell cycle core regulators might have been coopted to elicit additional functions in parallel to cell cycle control. We show that a positive cell cycle regulator, CDC25B, unexpectedly reduces proliferative divisions and promotes differentiation. Cell cycle regulators are routinely described as deregulated in cancers and are associated with increased proliferation. Understanding their function outside the cell cycle is therefore crucial to characterising their molecular and cellular mechanisms of action and to foresee novel therapeutic strategies.

## Materials and methods

### Embryos

Fertile chicken eggs at 38°C in a humidified incubator yielded appropriately staged embryos (*Hamburger and Hamilton, 1992*). Animal related procedures were performed according to EC guidelines (86/609/CEE), French Decree no. 97/748 and CNRS recommendations.

### Generating a *Cdc25b* floxed allele and a *Cdc25b* nesKO littermates

Experiments were performed in accordance with European Community guidelines regarding care and use of animals, agreement from the Ministère de l'Enseignement Supérieur et de la Recherche number: C3155511, reference 01024.01, and CNRS recommendations. To disrupt *Cdc25b* function, we generated a modified allele of *Cdc25b* (Mouse Clinical Institute, IGBMC, Illkirch). Using Homologous recombination in embryonic cells (ES), we inserted two LoxP sites, flanking exon 4 to exon 7 of the *Cdc25b* gene (referred to as Floxed allele). Upon Cre-mediated excision, exons 4 to 7 are deleted and following intron splicing, a premature stop codon is generated, leading to a truncated protein of 134 aa. The activity of this remaining peptide has been tested in a cellular model and has no activity (not shown). The mouse strain used is C57BL6/JRj. We first generated a mutant mouse line (*Cdc25b*$^{-/-}$) by crossing *Cdc25b* floxed mice with *PGK-Cre* mice, resulting in an ubiquitous and permanent deletion of *Cdc25b*. In order to delete *Cdc25b* activity specifically at the onset of neurogenesis, we crossed *Cdc25b*$^{fl/-}$ mice with transgenic mice expressing the Cre recombinase under the control of the rat *Nestin* (Nes) promoter and enhancer (*Tronche et al., 1999*). The effect of expressing Cre recombinase on proliferation and neurogenesis was evaluated by comparing *Cdc25b*$^{fl/+}$ and *NestinCre;Cdc25b*$^{fl/+}$ littermates. As there were no phenotypic differences between these embryos for any of the parameters that we measured (not shown), they were both included with the *Cdc25b*$^{fl/-}$ littermates in the control group.

### Statistical analysis of the mouse neural phenotype

For each experiment, at least three independent litters and three different slides per embryo were analyzed. To compare the number of neurons between control and conditional mutant embryos, we used a statistical model called the 'mixed effect model'. This model contains both the fixed effect, that is, the genotype of the embryo (control or conditional mutant) and random effects, that is, the variability induced by the age of the litter and by the embryo nested in the litter. Random effects were excluded using the R software and the package 'nlme', and we applied the following formula:

```
library(nlme)
result.lme <- lme(Neuronnumber ~Genotype,
        random =~1|Litter/Embryo, data = data, method=``REML'')
```

To test the effect of the genotype on the number of neuron, we next performed an ANOVA test. * p < 0.05; ** p < 0.01; *** p < 0.001.

## DNA constructs and in ovo electroporation

In ovo electroporation experiments were performed using 1.5- to 2-day-old chickens as described previously (*Peco et al., 2012*) . Loss of function was performed as described in (*Peco et al., 2012*). Gain-of-function experiments were performed using a vector expressing the various human CDC25 isoforms (hCDC25B3, hCDC25B3$^{\Delta CDK}$, hCDC25B3$^{\Delta P\ \Delta CDK}$) under the control of a cis regulatory element of the mouse Cdc25B called pccRE. A control vector was generated with the *β*Gal gene downstream of the pccRE. All gain-of-function experiments were performed at 1.5 µg/µl. For the Brainbow experiments, we used a pCX-Cre gift of X. Morin (*Morin et al., 2007*), at 0.5 ng/µl; Nucbow a gift of J. livet (*Loulier et al., 2014*) at 0.5 µg/µl. The Sox2p-GFP, Tis21p-RFP, and NeuroD-luciferase constructs were obtained from E. Marti and used at 1 µg/µl, 0.5 µg/µl and 1 µg/µl, respectively. pNLS EGFP-L2-PCNA was received from M.C. Cardoso (*Leonhardt et al., 2000*) and used at 0.5 µg/µl.

## In situ hybridization and immunohistochemistry on mouse and chick embryos

Mouse embryos were dissected in cold PBS and fixed in 4% paraformaldehyde overnight at 4°C. Then they were embedded in 5% low-melting agarose before sectioning on a Leica vibratome, in 50 µm thick transversal sections. In situ hybridization was performed as published (*Lacomme et al., 2012*). Riboprobes to detect *mCDC25B* transcripts were synthesized from linearized plasmid containing the full *CDC25B* cDNA. Riboprobe sequence : ACTCCTGTCGAAAGGGCTTCTGAAGAAGA TGACGGATTTGTGGACATCCTGGAGAGTGATTTAAAGGATGACGAGAAGGTCCCCGCGGGCA TGGAGAACCTCATTAGTGCCCCACTGGTCAAAAAGCTGGATAAGGAAGAGGAACAGGATCTCA TCATGTTCAGCAAGTGCCAGAGGCTCTTCCGCTCCCCATCCATGCCATGCAGTGTGA TCCGACCCATCCTCAAGAGGCTAGAGCGGCCCCAGGACCGGGATGTGCCTGTCCAGAG-CAAGCGCAGGAAAAGTGTGACACCCCTGGAAGAGCAGCAGCTTGAAGAACCTAAGGCCCGTG TCTTTCGCTCAAAGTCGCTGTGTCATGAGATTGAGAACATCCTGGATAGTGACCACCGTGGAC TGATCGGAGATTACTCTAAGGCCTTCCTCCTGCAGACCGTGGATGGCAAACACCAAGACCTTAAG TACATCTCACCAGAAACTATGGTGGCCCTGTTAACAGGCAAGTTCAGCAACATCGTGGAGAAA TTTGTCATTGTGGACTGCAGATACCCCTATGAGTATGAAGGCGGGCATATCAAGAATGCTG TGAACCTGCCCCTGGAACGGGATGCTGAGACCTTTCT. Immunohistochemistry was performed as described in (*Lobjois et al., 2004*). The antibodies used were the anti-Pax2 (Covance), guinea pig anti-Tlx3 (gift from C.Birchmeier, *Müller et al., 2005*, anti-Pax7 (Hybridoma Bank), and anti-Sox2 (Millipore). For chick embryos, proteins or transcripts were detected on 40 µm vibratome sections, as previously described (*Peco et al., 2012*). The antibodies used were: anti-HuC/D (Molecular Probes), anti-Sox2 (Chemicon), anti-PH3 (Upstate Biotechnology), anti-BrdU (mouse monoclonal, G3G4), anti-BrdU (rat anti-BrdU, AbD Serotec), anti-active caspase 3 (BD Biosciences), and anti-GFP (Invitrogen).

## Cell proliferation and survival analyses

Cell proliferation was evaluated by incorporation of 5-ethynyl-2'-deoxyuridine (Click-iT EdU Alexa Fluor 647 Imaging Kit, Invitrogen). 10 µl of 250 µM EdU solution were injected into chicken embryos harvested 30 min later, fixed for one hour and processed for vibratome sectioning. EdU immunodetection was performed according to manufacturer's instructions. Mitotic cells were detected using anti-PH3. G2-phase length was determined using the percentage of labeled mitoses (PLM) paradigm (*Quastler and Sherman, 1959*). EdU incorporation was performed as described above, except that a similar dose of EdU was added every 2 hr, and embryos were harvested from 30 to 180 min later. Embryos were fixed and labeled for both EdU and PH3. We then quantified the percentage of PH3 and EdU co-labeled nuclei with increasing times of exposure to EdU. The progression of this percentage is proportional to G2-phase duration. Cell death was analyzed by immunofluorescence, using the anti-active Caspase three monoclonal antibody (BD Biosciences).

### EdU incorporation in mice

For EdU staining experiments in mouse, 100 µl of 1 mg/ml EdU were injected intraperitoneally into pregnant mice. Litters were harvested 1, 2 or 3 hr following injection.

### Imaging and data analysis

Slices (40 µm) were analyzed using a SP5 Leica confocal microscope as described previously (*Peco et al., 2012*). Experiments were performed in triplicate. For each embryo, confocal analyses were performed on at least three slices. Confocal images were acquired throughout the slices at 3 µm z intervals.

### Tis21::RFP/Sox2::GFP Quantification

For each experimental slice, Z sections were acquired every 3 µm, and blind cell quantifications were performed on one out of every three Z sections to avoid counting the same cell twice. For each slice, the percentage of cells is determined using the sum of counted Z sections. For each experimental condition, the number of embryos analyzed and of cells counted is indicated in the Figure legend.

### In Vivo luciferase reporter assay

Embryos were electroporated with the DNAs indicated together with a NeuroDp-Luciferase reporter (*Saade et al., 2013*) and a renilla-construct (Promega) for normalization. GFP-positive neural tubes were dissected out at 48 hr after electroporation and homogenized in passive lysis buffer. Firefly- and renilla-luciferase activities were measured by the Dual Luciferase Reporter Assay System (Promega), and the data are represented as the mean ±sem from at least 14 embryos per experimental condition.

### Time-lapse imaging of cultured chick neural tube

1.5-days-old embryos were electroporated with a pNLS-EGFP-L2-PCNA (*Leonhardt et al., 2000*) vector, to distinguish the G2/M/G1 phases of the cell cycle, at 0.5 µg/µl. 6 hr later, embryos were dissected, fluorescent neural tubes were transferred to a tissue chopper (Mc Ilwain) and 100 µm thick transverse sections were sliced. Sections were collected in 199 culture medium (GIBCO) and were sorted out under a fluorescence microscope to control tissue integrity and the presence of isolated fluorescent cells along the dorso-ventral axis. Each slice was imbedded into 10 µl of rat type I collagen (Roche; diluted at 80% with 1X MEM (GIBCO), 1X GlutaMax (GIBCO) and neutralizing bicarbonate (GIBCO)). Four neural tube-containing collagen drops (5 µl) were distributed on a 35 mm glass-bottom culture dish (IBIDI). Collagen polymerization was performed at 38°C for 30 min and 1.5 ml of complete culture medium (199 medium, 5% FCS, 1X GlutaMax, Gentamicin 40 µg/ml) was gently added. For time-lapse, images were acquired on an inverted microscope (Leica inverted DMI8) equipped with a heating enclosure (set up at 39°C), a spinning disk confocal head (CSU-X1-M1N, Yokogawa) a SCMOS camera and a 63X oil immersion objective (NA 1,4–0,7). We recorded 40 µm thick z stacks (2 µm z-steps) at 5 min intervals. IMARIS and ImageJ software were used for image processing and data analysis.

### Statistics

Quantitative data are expressed as mean ± S.E.M. Statistical analysis was performed using the GraphPad Prism software. Significance was assessed by performing ANOVA followed by the Student- Mann-Whitney test, (*P<0.05, **P<0.01, ***P<0.001, ****P<0.0001 and n.s. non significant). See also Appendix 4.

## Acknowledgements

We are grateful to Dr. Elisa Marti for sharing plasmids. We thank Drs. Bertrand Bénazéraf, Alice Davy, Bernard Ducommun, Xavier Morin and Alain Vincent for critical reading of the manuscript and Dr. Caroline Monod for improving the English. We thank the CBI animal facilities and the Toulouse Regional Imaging platform (TRI) for technical support. We acknowledge the Developmental Studies Hybridoma Bank, created by the NICHD of the NIH and maintained at The University of Iowa, Department of Biology, Iowa City, IA 52242, for supplying monoclonal antibodies. Work in FP's

laboratory is supported by the Centre National de la Recherche Scientifique, Université P Sabatier, Ministère de L'Enseignement Supérieur et de la Recherche (MESR), the Fondation pour la Recherche sur le Cancer (ARC; PJA 20131200138) and the Fédération pour la Recherche sur le Cerveau (FRC; CBD_14-V5-14_FRC). Manon Azaïs, Fréderic Bonnet and Mélanie Roussat are recipients of MESR studentships. Angie Molina is a recipient of IDEX UNITI and Fondation ARC. The funding entities had no role in study design, data collection and analysis, decision to publish, or preparation of the manuscript.

## Additional information

### Funding

| Funder | Author |
|---|---|
| Centre National de la Recherche Scientifique | Frédéric Bonnet<br>Angie Molina<br>Mélanie Roussat<br>Manon Azais<br>Sophie Bel Vialar<br>Jacques Gautrais<br>Fabienne Pituello<br>Eric Agius |
| Ministère de l'Enseignement Supérieur et de la Recherche Scientifique | Frédéric Bonnet<br>Melanie Roussat<br>Manon Azais |
| Fondation ARC pour la Recherche sur le Cancer | Angie Molina<br>Fabienne Pituello |
| Université de Toulouse | Jacques Gautrais<br>Fabienne Pituello<br>Eric Agius |
| Fédération pour la Recherche sur le Cerveau | Fabienne Pituello |

The funders had no role in study design, data collection and interpretation, or the decision to submit the work for publication.

### Author contributions

Frédéric Bonnet, Data curation, Formal analysis, Investigation, Methodology; Angie Molina, Mélanie Roussat, Data curation, Formal analysis, Investigation; Manon Azais, Data curation, Investigation; Sophie Bel-Vialar, Conceptualization, Data curation, Formal analysis, Investigation, Writing—original draft; Jacques Gautrais, Conceptualization, Data curation, Formal analysis, Writing—original draft; Fabienne Pituello, Conceptualization, Data curation, Formal analysis, Funding acquisition, Validation, Investigation, Methodology, Writing—original draft, Project administration, Writing—review and editing; Eric Agius, Conceptualization, Data curation, Funding acquisition, Validation, Investigation, Methodology, Writing—original draft, Project administration, Writing—review and editing

### Author ORCIDs

Jacques Gautrais (ID) http://orcid.org/0000-0002-7002-9920
Eric Agius (ID) http://orcid.org/0000-0003-2123-9283

### Ethics

Animal experimentation: Experiments were performed in accordance with European Community guidelines regarding care and use of animals, agreement from the Ministère de l'Enseignement Supérieur et de la Recherche number: C3155511, reference 01024.01 and the CNRS recommendations.

### Decision letter and Author response

Decision letter https://doi.org/10.7554/eLife.32937.sa1

Author response https://doi.org/10.7554/eLife.32937.sa2

---

## Additional files

### Supplementary files

• Transparent reporting form

### Data availability

All data generated or analysed during this study are included in the manuscript and supporting files.

---

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

# Appendix 1

## Modeling the dynamics

In the appendices, we expose explicitly the hypotheses we made while interpreting the data of CDC25B experiments using a model of cell populations dynamics. In particular, we examine at the importance of a clear distinction between interpretations at the population level or at the cell level (*Altschuler and Wu, 2010* ). The model is first exposed at the population scale in Appendix 2. We derive an analytical solution when fate parameters are considered unvaried over the time window of the analyses. We show that the evolution is governed by one parameter: the balance between proliferation and differentiation. The model at the population scale can match many scenarios at the cell scale. In Appendix 3, we consider two contrasted scenarios that produce the same dynamics at the population scale. The first scenario considers that all cells divide at the same rate (all cells share a common cycle length), and that the choice by a cell to produce a symmetric proliferative division, a symmetric neurogenic division or an asymmetric division is stochastic. Such a stochastic choice at the cell scale is a very common interpretation for stem cells dynamics (*Harris, 1948*; *Loeffler and Roeder, 2004*; *Anderson, 2001*; *Antal and Krapivsky, 2010*; *Vogel et al., 1969*), even in the presence of (non-autonomous) external signaling (*Losick and Desplan, 2008*; *Johnston et al., 2007*; *Ramalho-Santos, 2004*. In this interpretation, the proportions of the modes of division at the population scale (the statistical measure over a large number of cells) is a direct reflection of the probabilities at the individual scale, provided all cells divide asynchronously with the same cycle length. The second scenario is used to test the opposite possibility: that the proportion of the modes of division at the population scale only comes from differences in cycle lengths, each mode of division having a specific cycle length.

We then present in Appendix 4 how our model statistics were used to enlighten the data. We point out that our model was not designed to "fit the data" by tuning free parameters, since it has no free parameters at all. It is used to check whether the modes of divisions (MoD) measured at HH17 were well in accordance with the neuronal fractions measured at HH22 given the measured cell cycle length, doing so with as few assumptions as possible.

Models for proliferation/differentiation in the spinal cord have been proposed previously in (*Saade et al., 2013* ) and (*Míguez, 2015*). However, a developmental switch has been incorporated in the first, which we do not use here. Also, we encounter a noteworthy difference between the second in (*Míguez, 2015*) and our model. In (*Míguez, 2015*), the model is built starting from cell description, and the limit to continuous-time population dynamics is taken considering that all mitoses are synchronous with a cell cycle length tending to 0 (which is implicit in equations 33-35 of (*Míguez, 2015*)-SI. Here, we first consider division rates at the population scale, and only then do we consider interpretations at the cell scale. Importantly, this difference between the two models yields different dynamics, especially when the balance between proliferation and differentiation of the progenitors is negative (i.e. in favor of differentiation). In the most extreme case (purely differentiating progenitors), our model still predicts the expected dynamics.

**eLife** Research article

Developmental Biology

## Appendix 2

### The model

We consider a population of cells $C(t)$ at time $t$, part of which are proliferating progenitors $P(t)$, part of which are differentiated neurons $N(t)$, with

$$C(t) = P(t) + N(t) \tag{4}$$

The dividing progenitors can undergo three kinds of fate, yielding:

- some proliferative divisions ending with two progenitors (pp-divisions)
- some asymmetric divisions ending with one progenitor and one neuron (pn-divisions)
- some terminal divisions ending with two neurons (nn-divisions)

We consider that the division of a cell in two cells is instantaneous (it is always possible to find a date before which there is one cell, and after which there are two cells).

We also consider that division events occur uniformly in time (asynchronously).

Let us denote :

$\eta$ the rate at which P-cells undergo divisions (in fraction of the P-pool per unit time)

$\alpha_{pp}(t)$ the fraction of dividing cells undergoing pp-divisions

$\alpha_{pn}(t)$ the fraction of dividing cells undergoing pn-divisions

$\alpha_{nn}(t)$ the fraction of dividing cells undergoing nn-divisions

$P(0), N(0)$ the quantity of P-cells and N-cells known at time $t = 0$.

In general, the fractions of pp-, pn- and nn-divisions can evolve with time, under the constraint that $\alpha_{pp} + \alpha_{pn} + \alpha_{nn} = 1$, and so might as well the division rate.

The time change $\dot{P}(t)$ of pool $P(t)$ (resp. $\dot{N}(t)$) is then driven at time $t$ by:

$$\begin{cases} \frac{dP}{dt} = \dot{P}(t) = -\eta P(t) + 2\alpha_{pp}(t)\eta P(t) + 1\alpha_{pn}(t)\eta P(t) \\ \frac{dN}{dt} = \dot{N}(t) = +2\alpha_{nn}(t)\eta P(t) + 1\alpha_{pn}(t)\eta P(t) \end{cases} \tag{5}$$

where in the first equation :

- $-\eta P(t)$ quantifies the rate at which P-cells disappear from the pool $P(t)$ because they divide. The quantity of disappearing P-cells between $t$ and $t + dt$ is then $\eta P(t)dt$
- $\alpha_{pp}\eta P(t)$ quantifies the fraction of this quantity that undergoes a pp-division ; it doubles to yield 2 P and adds up to the pool P(t) (hence the factor 2)
- $\alpha_{pn}\eta P(t)$ quantifies the fraction of this quantity that undergoes a pn-division ; it doubles to yield 1 P and 1 N, so only half (the P part) adds up to the pool P(t) (hence the factor 1)

correspondingly in the second equation :

- $\alpha_{nn}\eta P(t)$ quantifies the fraction of this quantity that undergoes a nn-division ; it doubles to yield 2 N and adds up to the pool N(t) (hence the factor 2)
- $\alpha_{pn}\eta P(t)$ is the fraction of this quantity that undergoes a pn-division ; it doubles to yield 1 P and 1 N and only half (the N part) adds up to the pool N(t) (hence the factor 1)

### Solutions with unvarying parameters

Considering a period of time during which the fractions of pp-, pn- and nn-divisions do not evolve with time, the dynamics can be written:

$$\begin{cases} \dot{P}(t) = -\eta P(t) + 2\alpha_{pp}\eta P(t) + 1\alpha_{pn}\eta P(t) \\ \dot{N}(t) = +2\alpha_{nn}\eta P(t) + 1\alpha_{pn}\eta P(t) \end{cases}$$

$$\begin{cases} \dot{P}(t) = (-1 + 2\alpha_{pp} + \alpha_{pn})\eta P(t) \\ \dot{N}(t) = (\alpha_{pn} + 2\alpha_{nn})\eta P(t) \end{cases} \tag{7}$$

Let $\gamma = -1 + 2\alpha_{pp} + \alpha_{pn}$.

Considering that $\alpha_{pp} + \alpha_{pn} + \alpha_{nn} = 1$, we have:

$$\begin{aligned}
\alpha_{pn} + 2\alpha_{nn} &= \alpha_{pn} + 2(1 - \alpha_{pp} - \alpha_{pn}) \\
&= \alpha_{pn} + 2 - 2\alpha_{pp} - 2\alpha_{pn} \\
&= 1 - (-1 + 2\alpha_{pp} + \alpha_{pn}) \\
&= 1 - \gamma
\end{aligned} \tag{8}$$

Hence,

$$\begin{cases} \dot{P}(t) &= \gamma\eta P(t) \\ \dot{N}(t) &= (1 - \gamma)\eta P(t) \end{cases} \tag{9}$$

and the solutions are of the general form:

$$\begin{cases} P(t) &= P(0)e^{\gamma\eta t} \\ N(t) &= N(0) + \int_0^t (1 - \gamma)\eta P(u)du \end{cases} \tag{10}$$

Plugging the first into the second, we have:

$$\begin{cases} P(t) &= P(0)e^{\gamma\eta t} \\ N(t) &= N(0) + (1 - \gamma)\eta P(0)\int_0^t e^{\gamma\eta u}du \end{cases} \tag{11}$$

## Explicit solutions

For explicit solutions, we have to consider two cases: $\gamma = 0$ and $\gamma \neq 0$.

For $\gamma = 0$, we have:

$$\begin{cases} P(t) &= P(0) \times 1 \\ N(t) &= N(0) + \eta P(0)\int_0^t 1 du \end{cases}$$

so that:

$$\begin{cases} P(t) &= P(0) \\ N(t) &= N(0) + \eta P(0)t \end{cases} \tag{12}$$

In that case, the pool of progenitors is steady, and the pool of neurons increases linearly with time.

For $\gamma \neq 0$, solving the integral in the second equation yields:

$$\begin{cases} P(t) &= P(0)e^{\gamma\eta t} \\ N(t) &= N(0) + (1 - \gamma)\eta P(0)\left(\frac{1}{\eta\gamma}(e^{\eta\gamma t} - e^{\eta\gamma 0})\right) \end{cases} \tag{13}$$

so that:

$$\begin{cases} P(t) &= P(0)e^{\gamma\eta t} \\ N(t) &= N(0) + P(0)\frac{1-\gamma}{\gamma}(e^{\eta\gamma t} - 1) \end{cases} \tag{14}$$

In that case, the evolution of the system depends on the sign of $\gamma$.

## Meaning of $\gamma$

We note that, for a given mitosis rate $\eta$, the dynamics only depend upon $\gamma$.

We have $\gamma = 2\alpha_{pp} + \alpha_{pn} - 1 = 2\alpha_{pp} + \alpha_{pn} - (\alpha_{pp} + \alpha_{pn} + \alpha_{nn}) = \alpha_{pp} - \alpha_{nn}$.

The case $\gamma = 0$ (*Equation 12*) corresponds to $\alpha_{pp} = \alpha_{nn}$. Here, the P-pool is steady and can be considered as a source of N-cells emitted at the steady rate $\eta P(0)$ (N-cells per unit time):

$$N(t) = N(0) + \eta P(0)t \ \text{(for } \alpha_{pp} = \alpha_{nn}) \tag{15}$$

The case $\alpha_{pp} > \alpha_{nn}$ yields $\gamma > 0$, so that the P-pool will increase with time. At the extreme, a purely proliferative P-pool corresponds to $\alpha_{pp} = 1$ and $\alpha_{nn} = 0$, hence $\gamma = 1$. In that case, the dynamics simplify to the classical proliferative equation for the P-pool, while the N-pool remains unchanged:

$$\begin{cases} P(t) & = P(0)e^{\eta t} \\ N(t) & = N(0) \end{cases} \text{(for } \alpha_{pp} = 1, \alpha_{nn} = 0) \tag{16}$$

The case $\alpha_{pp} < \alpha_{nn}$ yields $\gamma < 0$, so that the P-pool will decrease with time. At the extreme, a fully differentiating P-pool corresponds to $\alpha_{pp} = 0$ and $\alpha_{nn} = 1$, hence $\gamma = -1$. In that case, the P-pool undergoes a classical exponential decay, and the N-pool increases in proportion to the remaining P-pool, up to $2P(0)$:

$$\begin{cases} P(t) & = P(0)e^{-\eta t} \\ N(t) & = N(0) + P(0)(-2)(e^{-\eta t} - 1) \text{ (for } \alpha_{pp} = 0, \alpha_{nn} = 1) \\ & = N(0) + 2P(0)(1 - e^{-\eta t}) \end{cases} \tag{17}$$

Regarding the total population $C(t) = P(t) + N(t)$ (**Appendix 2—Figure 1**), positive (or null) value of $\gamma$ ($\alpha_{pp} \geq \alpha_{nn}$) allows an infinite growth of the total population $C(t)$ whereas the growth saturates as soon as $\gamma < 0$ ($\alpha_{pp} < \alpha_{nn}$). Since we made the hypothesis that the fate parameters were considered as steady over time, interpretations for the real biological system should take into account that these fate parameters actually change over longer time in the real system.

Regarding the fraction of neurons in the population, $N(t)/C(t)$ (**Appendix 2—Figure 2**), it increases as soon as $\gamma < 1$, yet at a rate depending on $\gamma$.

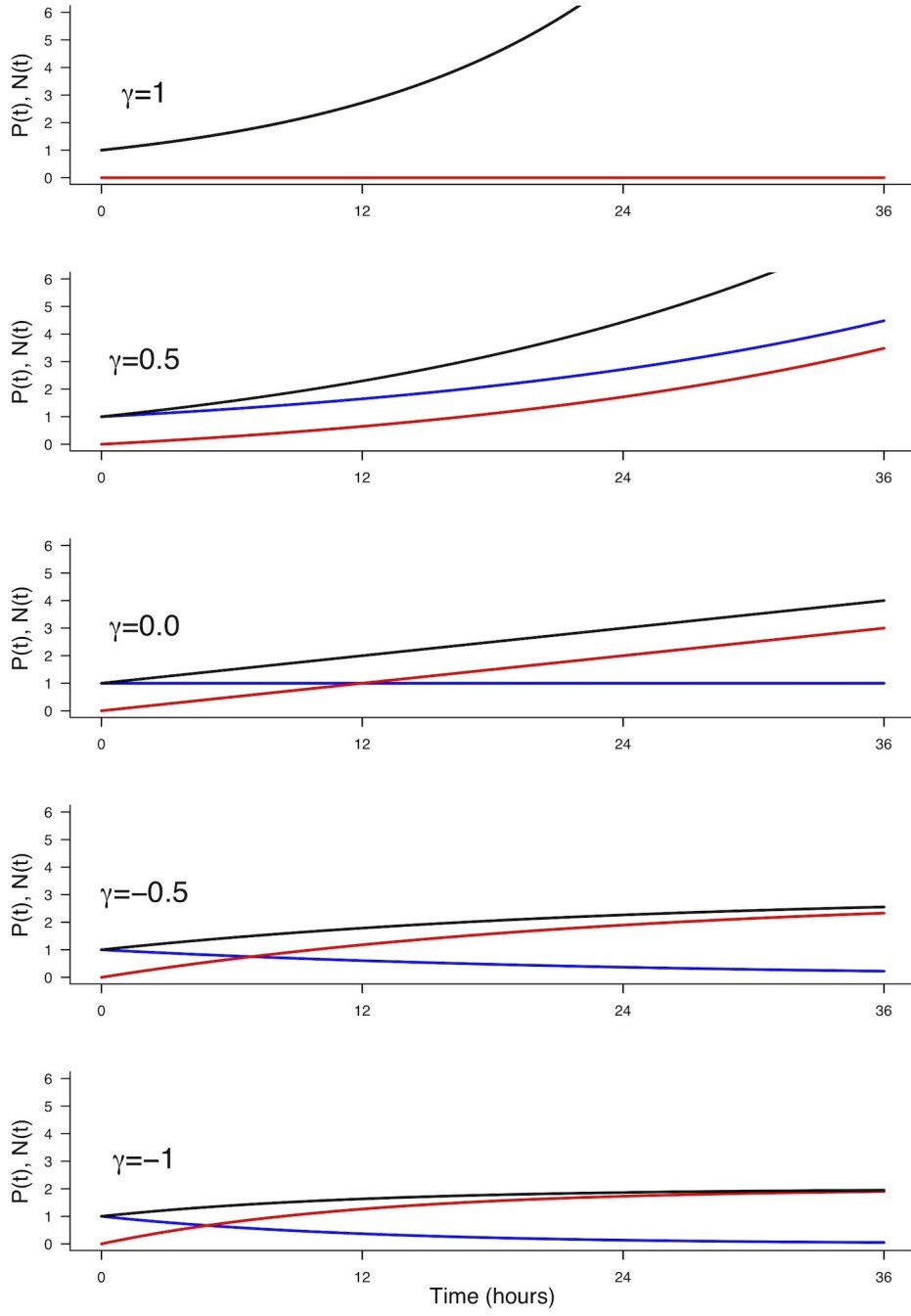

**Appendix 2—figure 1.** Effect of $\gamma$ on the evolution of $P(t)$ (blue), $N(t)$ (red) and $C(t) = P(t) + N(t)$ (black). Parameters used: $P(0) = 1$, $N(0) = 0$, $\eta = 1/12$, corresponding to a cycle time of 12 hr.

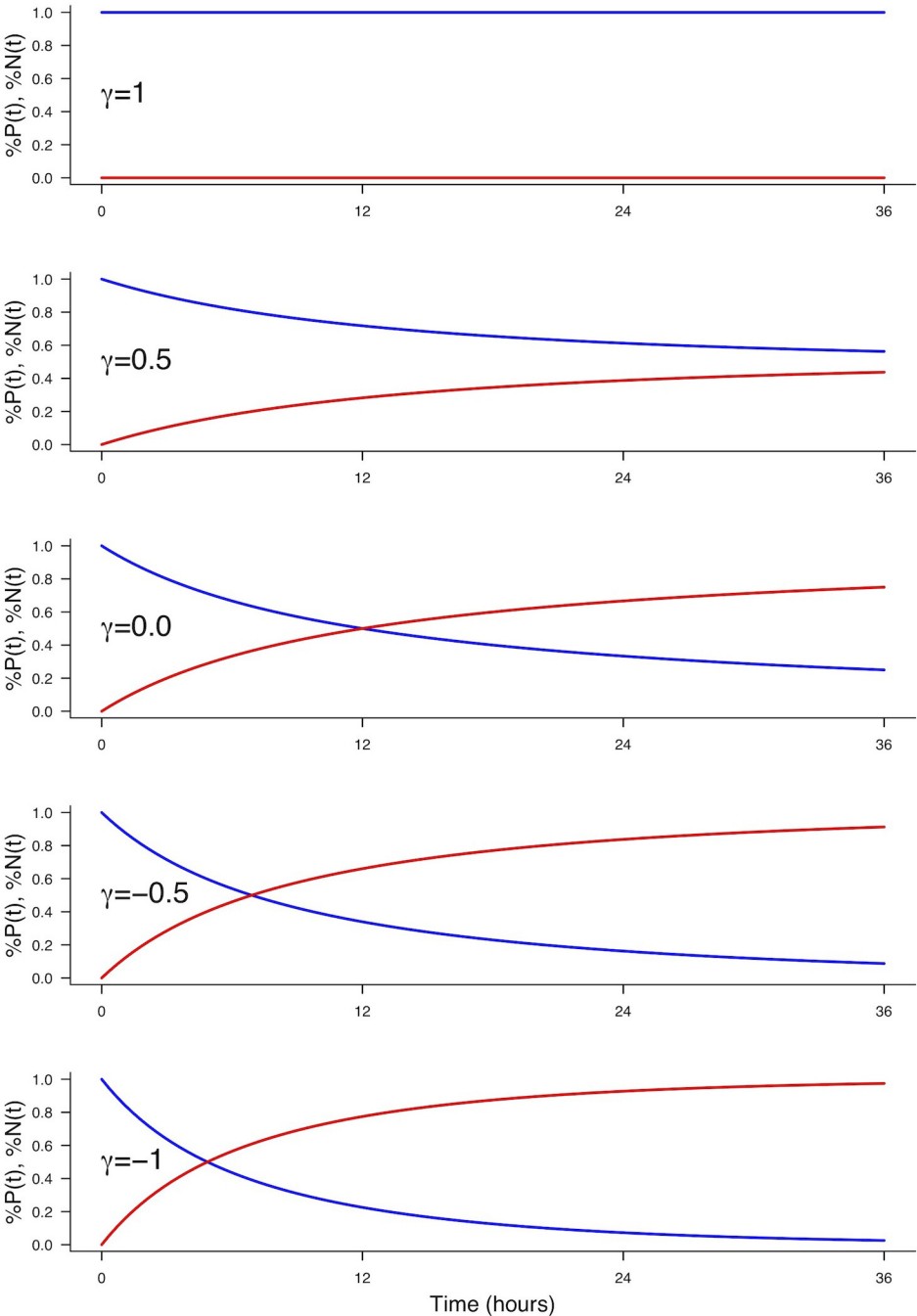

**Appendix 2—figure 2.** Effect of $\gamma$ on the evolution of the fractions $P(t)/C(t)$ (blue) and $N(t)/C(t)$ (red).

# Appendix 3

## Interpretations at the individual cell scale

We have so far described the system at the population scale. At the individual scale, two different kinds of process (at least) would result in the same dynamics at the population scale described in *Equation 5*.

## Probabilistic fates, with a common deterministic division rate

The most immediate interpretation is to consider that all cells undergo mitosis at the same rate, and that the fate of any mitosis is stochastic and probabilistically distributed according to $(\alpha_{pp}, \alpha_{pn}, \alpha_{nn})$. In that case, only the rate $\eta$ (used in the equations at the population scale) has to be determined from a cell-scale model, since it depends upon the characteristic time $\tau_m$ between two mitoses at the cell scale.

Let us consider the hypothesis that mitosis happen exactly every $\tau_m$ for all cells (common deterministic division time), still asynchronously so that division dates are uniformly distributed over time (this is the most common hypothesis in the community). We want to express $\eta$ as a function of $\tau_m$.

For the sake of simplicity, let us consider the pure proliferative process ($\alpha_{pp} = 1$) so that we deal with only one population $P(t)$.

Let us start at time 0 with an initial pool $P_1(0)$ containing a very large number of cells (so that $P_1(t)$ can be considered as continuous). Since mitoses take a fixed time $\tau_m$, their last division occurred before $t = 0$, the oldest division happened at $0 - \tau_m$ and they all will undertake a mitosis in the time interval $[0 .. 0 + \tau_m]$. Since divisions are uniformly distributed over time, the number performing a mitosis during a small time interval $\Delta t$ is proportional to $\Delta t / \tau_m$ and $P(0)$. Hence, the loss in $P_1$ between $t$ and $t + \Delta t$ is given by:

$$P_1(t + \Delta t) - P_1(t) = -P_1(0)\Delta t / \tau_m \tag{18}$$

$$\frac{P_1(t + \Delta t) - P_1(t)}{\Delta t} = -P_1(0)/\tau_m \tag{19}$$

Taking the limit $\Delta t \to 0$ yields:

$$\dot{P}(t) = \frac{dP_1(t)}{dt} = -P_1(0)/\tau_m \tag{20}$$

Considering $P_1(0)$, we then have:

$$\begin{aligned} P_1(t) &= P_1(0) - (P_1(0)/\tau_m)\, t \\ &= P_1(0)(1 - t/\tau_m) \end{aligned} \tag{21}$$

Logically, $P_1(t)$ decreases linearly from $P_1(0)$ down to 0 at time $t = \tau_m$. Meanwhile, the output of each division will populate the next generation, say $P_2(t)$, at twice the rate $P_1$ disappears, up to $2P_1(0)$ at time $t = \tau_m$, from which $P_2$ will start decreasing doing mitosis and populate the third generation $P_3$ and so on… Such a process would then translate into a population growth which is piecewise linear (*Appendix 3—figure 1*), but very close to an exponential growth. If we equate at time $\tau_m$ the piecewise growth, and its exponential approximation at rate $\eta$, we have:

$$e^{\eta\tau_m} = P_2(\tau_m) = 2\eta = \ln 2/\tau_m \tag{22}$$

Denoting $\tau_c = 1/\eta$ the characteristic time at the population scale, we then have: $\tau_c = \tau_m/\ln 2$. Hence, from an observed time $\tau_c$ at the population scale, we should infer (under this model) that $\tau_m = \tau_c \ln 2$, that is $\tau_m \simeq 0.7\tau_c$ (e.g. if population cycle time is 12 hr, cell cycle time should be around 8h20).

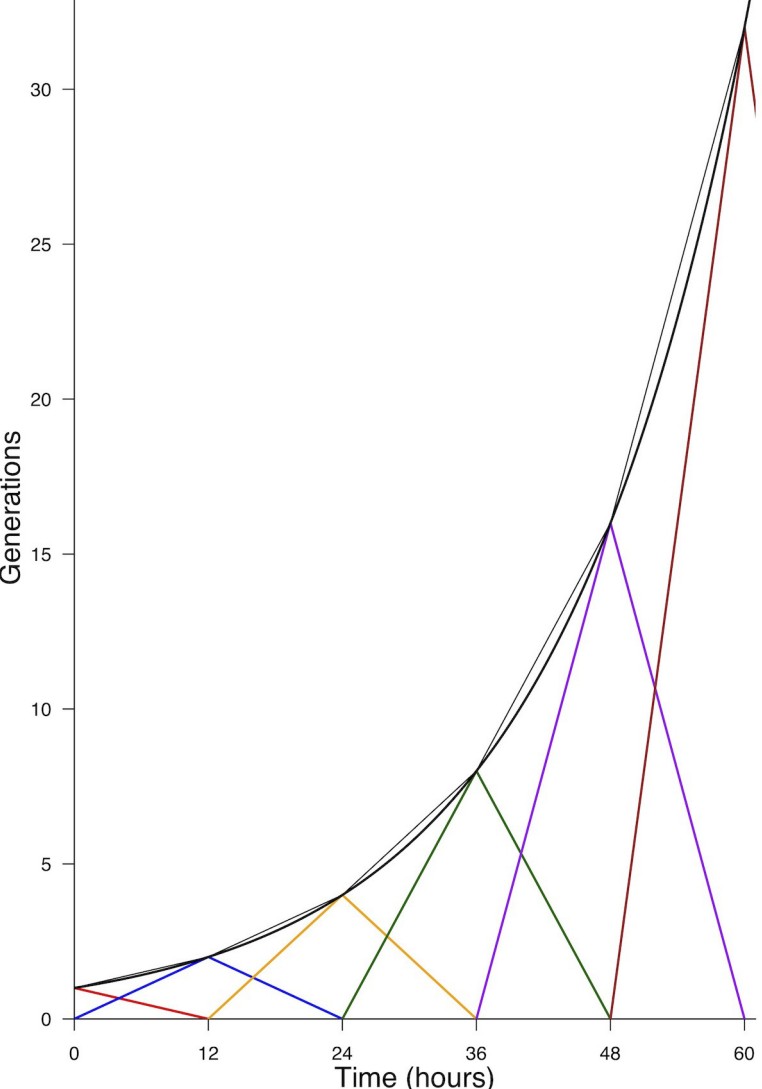

**Appendix 3—figure 1.** Generations produced by an initial pool. $P_1(0) = 1$, under the hypothesis of a common deterministic division time $\tau_m = 12$ h. Each generation is reported by a color. The thin black curve indicates the total pool present at time $t$ (adding the two generations). The thick black curve reports the continuous approximation $\exp(\ln 2\ t/\tau_m)$ (*Equation 22*).

## Deterministic fates, with specific division rates

Another way to produce the dynamics described in *Equation 5* at the population scale is to consider that each kind of fate result from a specific division time. In such a picture, the time needed to achieve a cycle deterministically determines the kind of fate.

To exhibit this interpretation, we rewrite *Equation 5* as follows:

$$\begin{cases} \dot{P}(t) & = -\eta(\alpha_{pp} + \alpha_{pn} + \alpha_{nn})P(t) + 2\alpha_{pp}\eta P(t) + 1\alpha_{pn}\eta P(t) \\ \dot{N}(t) & = 1\alpha_{pn}\eta P(t) + 2\alpha_{nn}\eta P(t) \end{cases} \tag{23}$$

Denoting $\eta_{pp} = \alpha_{pp}\eta$ (and correspondingly for $\eta_{pn}$ and $\eta_{nn}$), we then have:

$$\begin{cases} \dot{P}(t) & = -(\eta_{pp} + \eta_{pn} + \eta_{nn})P(t) + 2\eta_{pp}P(t) + 1\eta_{pn}P(t) \\ \dot{N}(t) & = 1\eta_{pn}P(t) + 2\eta_{nn}P(t) \end{cases} \tag{24}$$

The interpretation is then that, from the pool P(t), the cells leaving it at rate $\eta_{pp}$ yield pp-divisions,

those leaving it at rate $\eta_{pn}$ yield pn-divisions, and the others, leaving it at rate $\eta_{nn}$, yield nn-divisions. Overall, the pool $P(t)$ depletes at the sum rate $\eta = \eta_{pp} + \eta_{pn} + \eta_{nn}$.

Correspondingly, the population cycle time $\tau_c = 1/\eta$ would then be given by:

$$\frac{1}{\tau_c} = \frac{1}{\tau_{pp}} + \frac{1}{\tau_{pn}} + \frac{1}{\tau_{nn}} \tag{25}$$

equivalently by:

$$\tau_c = \frac{\tau_{pp}\tau_{pn}\tau_{nn}}{\tau_{pn}\tau_{nn} + \tau_{pp}\tau_{nn} + \tau_{pp}\tau_{pn}} \tag{26}$$

We also note that the distribution of fates is then completely constrained by the $\tau_{pp}, \tau_{pn}, \tau_{nn}$ (under the constraint that mitosis events are uniformly distributed in time). Indeed, it remains true that the quantity leaving the P-pool during $\Delta t$ to make pp-divisions is proportional to $\Delta t / \tau_{pp}$ (corr. for other fates). This implies in turn that the fraction $\alpha_{pp}$ leaving for an pp-division is $\tau_c / \tau_{pp}$, correspondingly, $\alpha_{pn} = \tau_c / \tau_{pn}$ and $\alpha_{nn} = \tau_c / \tau_{nn}$.

As a consequence, if we have experimental measures of $\tau_c$ and of a distribution among fates $\alpha_{pp}, \alpha_{pn}, \alpha_{nn}$, we must conclude that:

$$\tau_{pp} = \frac{\tau_c}{\alpha_{pp}}, \ \tau_{pn} = \frac{\tau_c}{\alpha_{pn}}, \ \tau_{nn} = \frac{\tau_c}{\alpha_{nn}} \tag{27}$$

For $\tau_c = 12\,h$, and a distribution $(0.6, 0.3, 0.1)$, we would obtain:

$$\tau_{pp} = 20\,h, \ \tau_{pn} = 40\,h, \ \tau_{nn} = 120\,h \tag{28}$$

The main point is then: if the ratios between fractions of fate $\alpha_{pp}, \alpha_{pn}, \alpha_{nn}$ resulted only from differences in rates $\eta_{pp}, \eta_{pn}, \eta_{nn}$, the ratios between rates must be the same as the ratios between fractions:

$$\frac{\eta_{pp}}{\eta_{nn}} = \frac{\alpha_{pp}}{\alpha_{nn}} \ ; \ \frac{\eta_{pp}}{\eta_{pn}} = \frac{\alpha_{pp}}{\alpha_{pn}} \ ; \ \frac{\eta_{pn}}{\eta_{nn}} = \frac{\alpha_{pn}}{\alpha_{nn}} \tag{29}$$

With $\alpha_{pp} = 0.6$, $\alpha_{nn} = 0.1$, we would have $\tau_{nn} = (\alpha_{pp}/\alpha_{nn})\tau_{pp} = 6\,\tau_{pp}$.

If we exclude the possibility that a nn-division is six times as long as a pp-division, then the distribution of fates can not be exclusively determined by differences in fate-based cycle times. It does not exclude that a given kind of fate (e.g. proliferative divisions pp) would require a longer time to be achieved than others, it excludes that such differences would suffice per se to explain the differences between the fractions of fates.

## Appendix 4

### Model predictions using (noisy) data

We obtain experimental measures with this system at different times after electroporation (time 0 hr): the fractions $f_N(24)$ of neurons at 24 hr and $f_N(48)$ at 48 hr (the fraction among the electroporated cells), the distribution of fates at 24 hr as well as an estimate of $\tau_c = 12$ hours. We hypothesize that the fate distribution is steady between 24 hr and 48 hr after electroporation, that is the 24 hr between quantification of the mode of division and progenitors and neuron counting. We use the model to check the consistency of these data with the model.

### Knowing the fractions of neurons at 24 h and 48 h, confidence intervals upon the fate distribution

The first test of consistency was to determine the range of distribution of fates which was able to explain the transition from $f_N(24)$ to $f_N(48)$.

If we had a system with only symmetric divisions (e.g. some value for $\alpha_{pp}$, $\alpha_{nn} = 1 - \alpha_{pp}$, with $\alpha_{pn} = 0$), we first ensured that one pair $(f_N(24), f_N(48))$ would be compatible with only one fate distribution.

Considering $P(24) + N(24) = 1$ arbitrary total amount of cells at 24 hr, we can plug $N(24) = f_N(24)$ and $P(24) = 1 - f_N(24)$ into *Equation 14* and get:

$$\begin{cases} P(48) &= (1 - f_N(24))e^{24\gamma\eta} \\ N(48) &= f_N(24) + (1 - f_N(24))\frac{1-\gamma}{\gamma}(e^{24\eta\gamma} - 1) \end{cases} \tag{30}$$

where $P(48), N(48)$ correspond to the amount obtained at 48 hr from this arbitrary amount of 1 at 24 hr. We have $f_N(48) = N(48)/(N(48) + P(48))$, yielding :

$$f_N(48) = \frac{\left[ f_N(24) + (1 - f_N(24))\frac{1-\gamma}{\gamma}(e^{24\eta\gamma} - 1) \right]}{\left[ f_N(24) + (1 - f_N(24))\frac{1-\gamma}{\gamma}(e^{24\eta\gamma} - 1) \right] + \left[ (1 - f_N(24))e^{24\gamma\eta} \right]} \tag{31}$$

which holds for any initial cell amount (*Appendix 4—figure 1*).

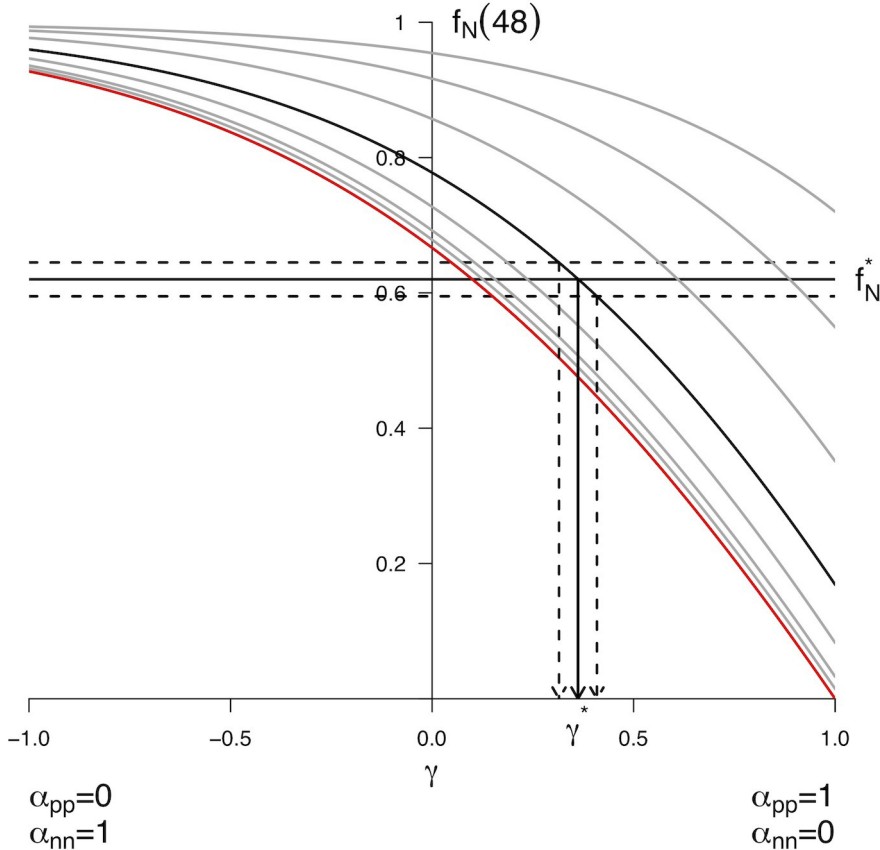

**Appendix 4—figure 1.** Predicted $f_N(48)$ from $f_N(24)$ for every distribution of symmetric division. The different curves correspond to different starting values $f_N(24)$ taken in $(0.0, 0.1, 0.2, 0.4, 0.6, 0.8, 0.9, 0.95)$. The bold line corresponds to $f_N(24) = 0.6$, the red line to $f_N(24) = 0.0$. Each curve reports the predicted value for $f_N(48)$ starting from the corresponding $f_N(24)$, and for all possible distributions of fates given by $\gamma = \alpha_{pp} - \alpha_{nn}$ (x-axis). Each combined $(f_N(24), \gamma)$ yields only one predicted $f_N(48)$. Conversely, experimental values for the pair $(f_N(24), f_N(48))$ allow to retrieve the corresponding $\gamma$ theoretical value. As an example, the value corresponding to the arbitrary value $f_N^* = 0.62$ was retrieved numerically using *Equation 31*. We found $\gamma^* = 0.362$, yielding $\alpha_{pp} = 0.681$ and $\alpha_{nn} = 0.319$. Confidence interval upon the distributions of fates can also be drawn using the experimental noise about $f_N(48)$, as illustrated here considering $f_N^* \pm 2.5\%$.

Now considering the full system with the three kinds of division, there is more than one unique triplet $(\alpha_{pp}, \alpha_{pn}, \alpha_{pn})$ that is compatible with the unique value of observed $(f_N(24), f_N(48))$. For instance, less nn-divisions can be compensated for by more pn-divisions, yielding the same $f_N(48)$.

We used the model in the same spirit as in *Appendix 4—figure 1* to compute the predicted values for $f_N(48)$ for all possible fate triplets. For the system with symmetric-only divisions above, the space of parameters for division is one-dimensional: $\gamma$ corresponds to one value of $\alpha_{pp}$, which constrains in turn the value of $\alpha_{nn}$. With the three kinds of division, this space of parameters becomes two-dimensional: we need to fix $\alpha_{pp}$ and $\alpha_{nn}$, and $\alpha_{pn}$ is then constrained. Hence the predictions should be drawn over a two-dimensional map.

We compute those maps for each experimental condition, starting from the corresponding observed value $f_N(24)$ (fixing the observed initial condition corresponds here to drawing only the bold curve in *Appendix 4—figure 1*). Then, we determine numerically the subset of fate triplets compatible with the $f_N(48) = f_N^*$ measured in the condition. We also determined numerically the confidence regions for the distributions of fates that can yield $f_N^* \pm 2.5\%$, $f_N^* \pm 5\%$ and $f_N^* \pm 10\%$.

In the end, we also report the distribution of fates that was actually measured in each condition, and check in which confidence interval it is (Ventral zone: *Appendix 4—figure 2*, *Appendix 4—*

*figure 3, Appendix 4—figure 4,* Dorsal zone: *Appendix 4—figure 5, Appendix 4—figure 6, Appendix 4—figure 7).*

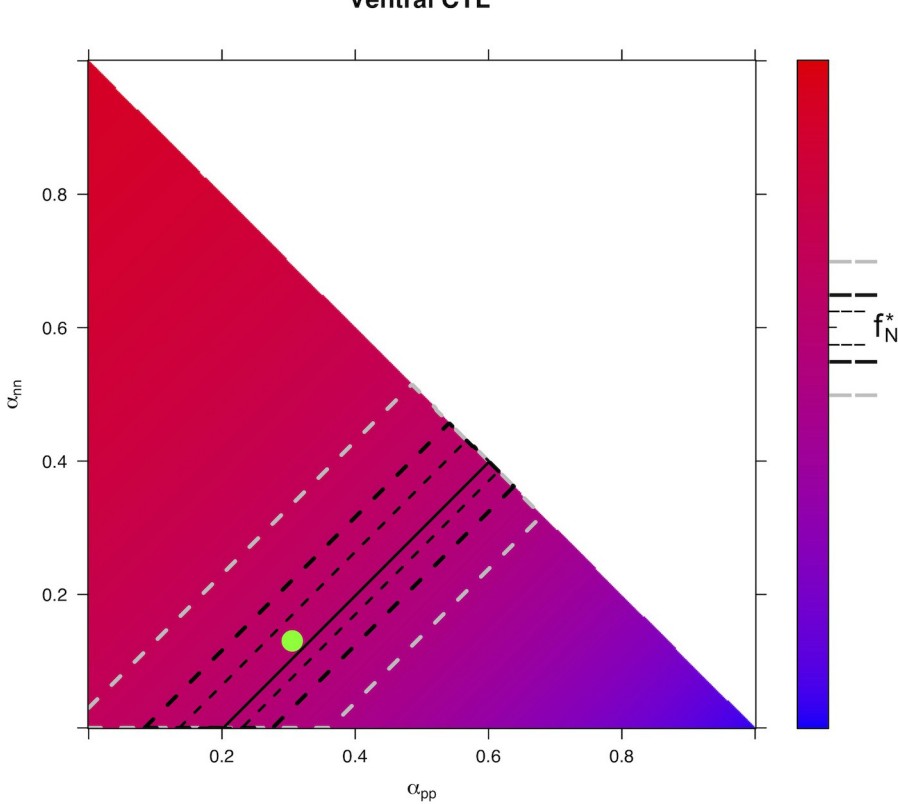

**Ventral CTL**

**Appendix 4—figure 2.** Predicted $f_N(48)$ from $f_N(24)$ for every distribution of fates for control condition in Ventral area. The color scale indicates $f_N(48)$. It is computed from the model, starting from the experimental value of $f_N(24)$ in the prevailing condition, and using all possible distributions of fates $\alpha_{pp}$ (x-axis), $\alpha_{nn}$ (y-axis) and $\alpha_{pn} = 1 - \alpha_{pn} - \alpha_{nn}$. The upper side of the triangle corresponds to $\alpha_{pn} = 0$. Confidence interval upon the predicted distributions of fates are drawn for the experimental value $f_N(48) = f_N^*$. Plain line: all distributions of fates giving exactly $f_N^*$. Region delimited by thin dotted line: all distributions of fates compatible with $f_N^* \pm 2.5\%$, thick dotted line : $f_N^* \pm 5\%$, gray dotted line: $f_N^* \pm 10\%$. Green dot: observed distribution of fates.

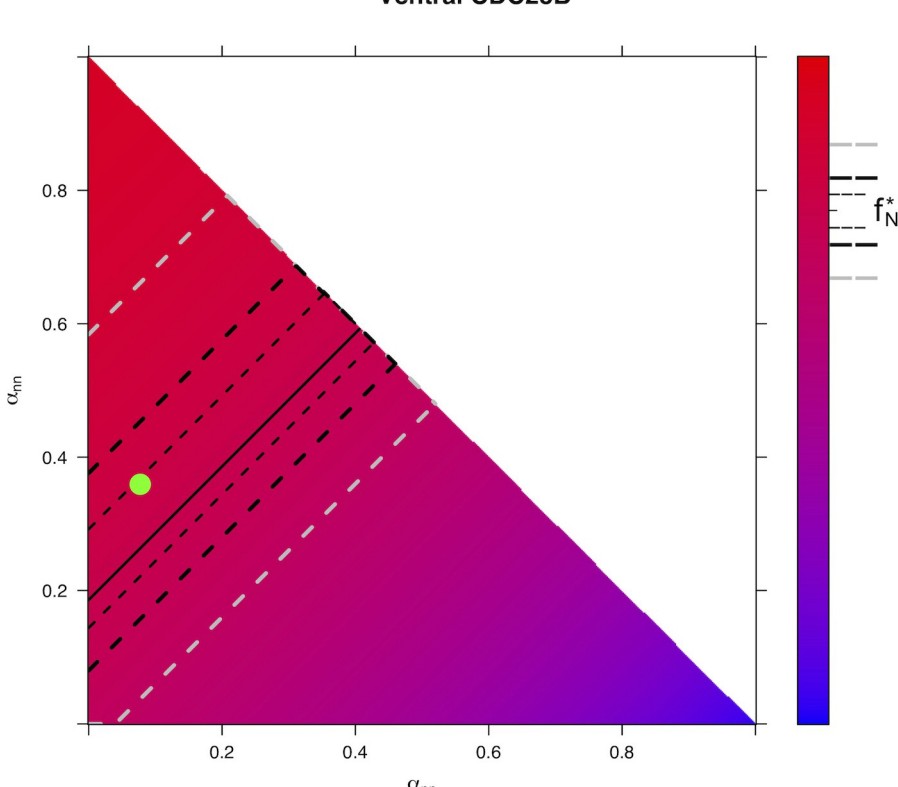

**Appendix 4—figure 3.** Predicted $f_N(48)$ from $f_N(24)$ for every distribution of fates for $\mathrm{CDC25B}$ condition in Ventral area.

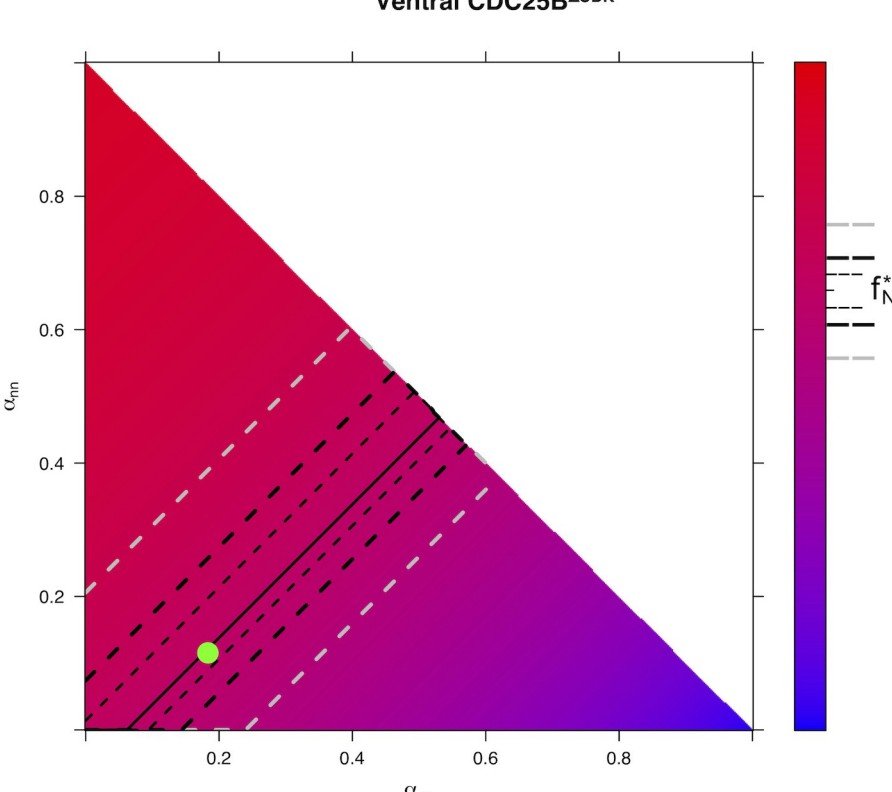

**Appendix 4—figure 4.** Predicted $f_N(48)$ from $f_N(24)$ for every distribution of fates for $CDC25B^{\Delta CDK}$ condition in Ventral area.

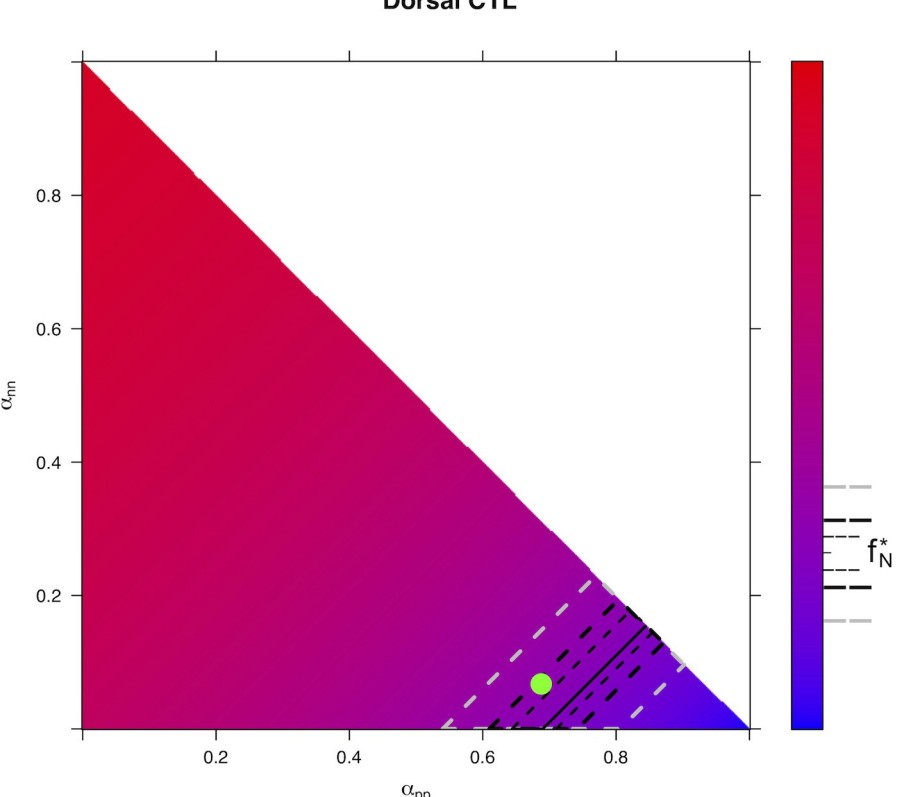

**Appendix 4—figure 5.** Predicted $f_N(48)$ from $f_N(24)$ for every distribution of fates for control condition in Dorsal area.

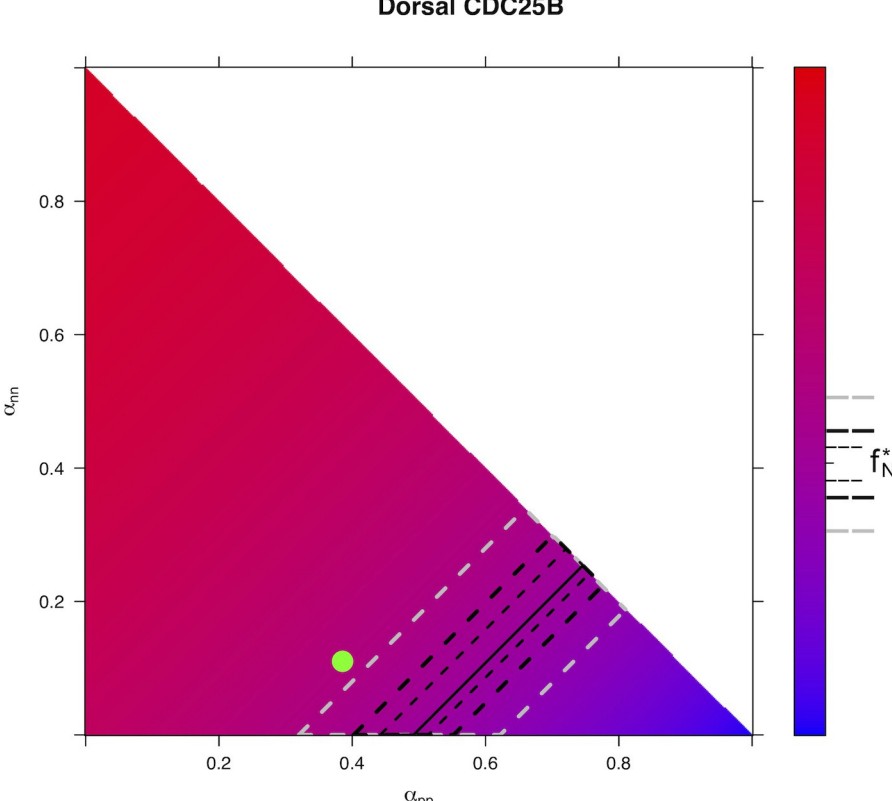

**Appendix 4—figure 6.** Predicted $f_N(48)$ from $f_N(24)$ for every distribution of fates for $\mathrm{CDC25B}$ condition in Dorsal area.

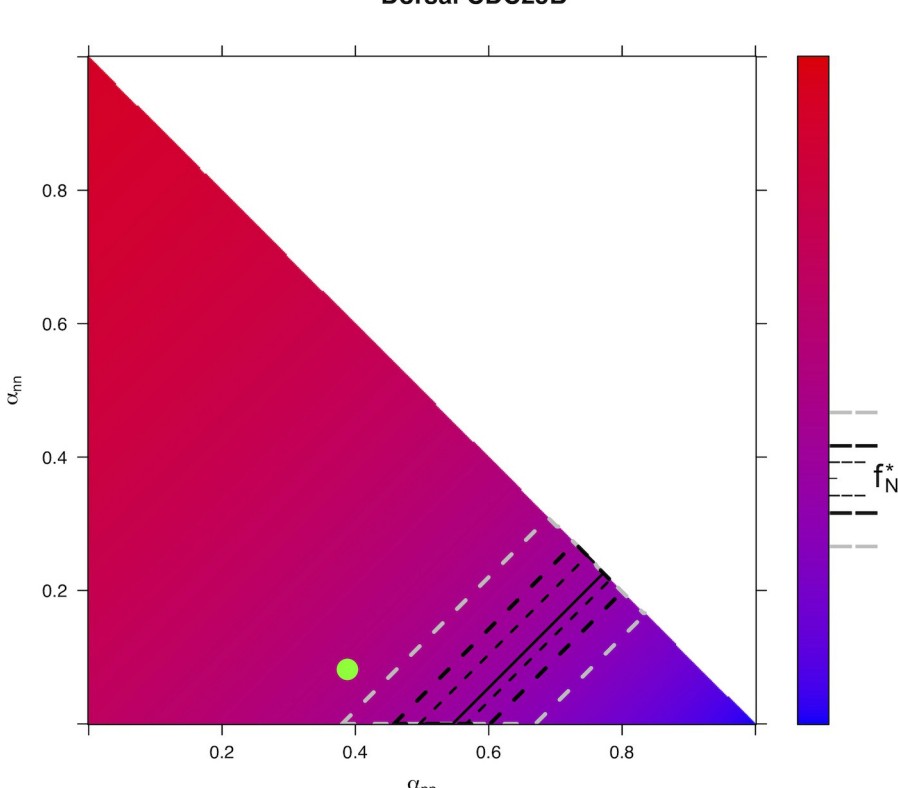

**Appendix 4—figure 7.** Predicted $f_N(48)$ from $f_N(24)$ for every distribution of fates for $\text{CDC25B}^{\Delta\text{CDK}}$ condition in Dorsal area.

## Predicted fraction of neurons at 48 hr knowing the fractions of neurons and the fate distribution at 24 hr

To compute the predicted fractions of neurons at 48 hr (after electroporation) reported in the main text (*Figure 6C*), we used *Equation 31*, parametrized by the data obtained for the averaged fraction of neurons at 24 hr (a.e.), the fate distribution at 24 hr (a.e.), and the cell cycle 12 hr.

All predictions are gathered in *Appendix 4—figure 8* as a function of the change in the proliferation/differentiation balance of the progenitors, induced by the CDC25B and the $\text{CDC25B}^{\Delta\text{CDK}}$ experiments. Together, the observations indicate that CDC25B and $\text{CDC25B}^{\Delta\text{CDK}}$ result in an increased proportion of neurons 48 hr a.e. (HH22).

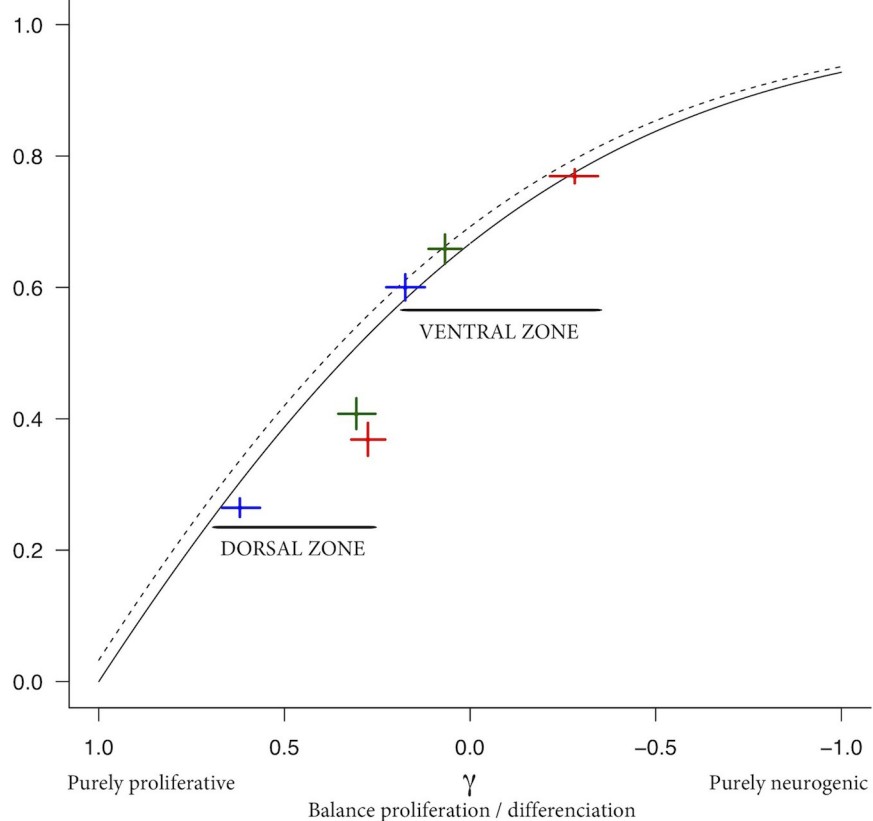

**Appendix 4—figure 8.** Predicted $f_N(48)$ from $f_N(24)$ varying the balance proliferation/differentiation $\gamma$. Plain line reports the model prediction for the dorsal zone, dotted line the model prediction for the ventral zone (predictions differ due to differences in the initial fraction $f_N(24)$ in the two zones). The experimental data are reported by crosses (cross arm lengths are 95% CI). Blue cross: CTL, red cross: CDC25B, green cross: CDC25B$^{\Delta CDK}$.

Such an increased *proportion* of neurons is actually compatible with two dynamical scenarios regarding how the absolute amounts of the two pools (progenitors, neurons) are modified by CDC25B gain-of-function: scenario (1) a speed-up of the neuron pool so that it increases faster under the gain-of-function at the expense of the progenitor pool expansion, or scenario (2) a decrease of the progenitor pool while the pool of neurons keeps the same expansion rate. Which scenario is relevant depends on how CDC25B affects the balance $\gamma$ between proliferation and differentiation.

The progenitor pool can increase only if $\gamma>0$, which implies $\alpha_{pp}>\alpha_{nn}$. In this case, the two pools can increase (scenario 1), their respective growth rates are controlled by $\gamma$ and the neurogenic effect of CDC25B gain-of-function will produce a greater absolute number of neurons in the end (at 48h/HH22). Otherwise ($\gamma<0$, that is $\alpha_{pp}<\alpha_{nn}$), the neuron pool can increase at about the same rate, yielding the same absolute number of neurons at 48 hr/HH22, and the increased fraction of neurons reflects a depletion of the progenitor pool (scenario 2).

The model enlightens which is the most probable scenario for the dynamical impact of CDC25B manipulation, since we can compute the underlying evolution of the absolute amounts of the two pools that determines the evolution of the neuronal fraction (*Appendix 4 — figure9C*).

Under CDC25B gain-of-function in the dorsal neural tube (*Appendix 4 — figure9C*-right), the percentage of progenitors performing pp-divisions stays greater than the percentage of those performing nn-divisions ($38.6\%>11.3\%$, $\alpha_{pp}>\alpha_{nn}$) and the balance is still positive ($\gamma = 0.386 - 0.113 = 0.273>0$), so the pool of progenitors still increases but at a lower rate than the control (where $\gamma = 0.663 - 0.078 = 0.585$). The higher percentage of neurons at 48 hr/HH22 then results from an even higher absolute number of neurons (scenario 1).

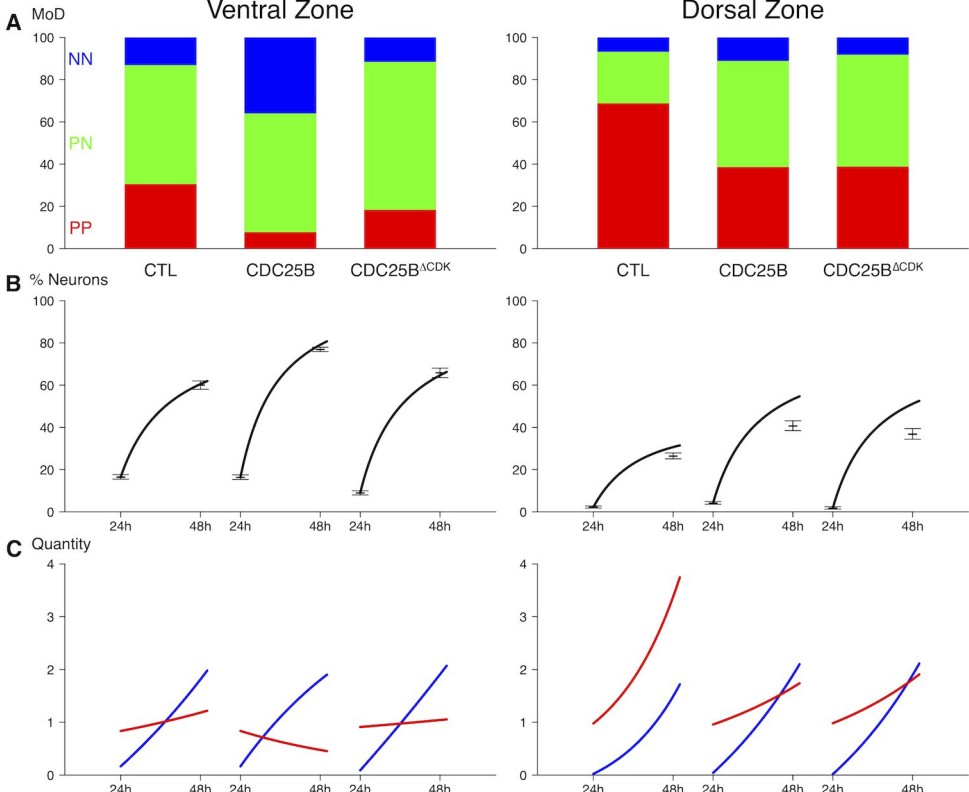

**Appendix 4—figure 9.** Summary of the data and predictions. (**A**) Observed distributions of modes of divisions (MoD) for the three conditions and the two zones. (**B**) Predicted evolutions of the neuronal fraction from $f_N(24)$ to $f_N(48)$ given the observed distribution of fates (lines) and observed fractions at 24 hr and 48 hr. (**C**) Corresponding evolution in numbers of the two pools (Red: progenitors, Blue: neurons).

By contrast, in the ventral neural tube, the balance shifts from $\gamma = 0.393 - 0.127 = 0.266$ in the control to $\gamma = 0.069 - 0.407 = -0.338$, becoming negative under $CDC25B$ gain-of-function (scenario 2). Accordingly, the absolute number of neurons at 48 hr/HH22 is poorly affected, but the pool of progenitors declines, explaining the higher fraction of neurons (*Appendix 4 — figure9C*-left).

