## [Decision Letter]

Thank you for sending your article entitled "Neurogenic decisions require a cell cycle independent function of the CDC25B phosphatase" for peer review at *eLife*. Your article is has been evaluated by two peer reviewers, and the evaluation has been overseen by a Reviewing Editor and Didier Stainier as the Senior Editor.

The major issue to be addressed is the non-cell cycle function of CDC25B. The data presented are intriguing and indicate an alternative role for this protein. Some data regarding the nature of that role would be required for acceptance to *eLife*.

*Reviewer #1:*

Bonnet et al. describe a novel function of Cdc25B in promotion of neurogenesis in the developing mouse and chick spinal cord. They first confirmed in mice their previous results obtained with chick embryos showing that Cdc25B ablation prolongs G2 phase of the cell cycle and attenuates neurogenesis. Conditional knockout of Cdc25B reduced the numbers of both Pax2+ GABAergic neurons and Tlx3+ glutamatergic neurons, without significantly affecting that of Pax7+ progenitors. With the use of a cell cycle-dependent cis element, the authors also showed that overexpression of Cdc25B is sufficient to promote neuronal differentiation in the chick spinal cord without affecting orientation of the mitotic spindle or asymmetry of spindle size. Furthermore, Cdc25B depletion and overexpression reduced and increased Tis21+ neurogenic divisions, respectively. Interestingly, this neurogenic function of Cdc25B appears to be independent of its cell cycle-regulating activity, given that overexpression of a Cdc25B mutant that lacks the CDK binding domain still promotes neuronal differentiation. A mathematical model supports the notion that modification of cell cycle duration does not simply account for changes in the rate of neurogenic division under given conditions.

The study was well executed and the manuscript is well written. It would greatly strengthen the study if the authors could show the nature of the cell cycle-independent mechanism by which Cdc25B promotes neurogenic division, but this may be too much to ask.

*Reviewer #2:*

This manuscript considers the role of CDC25B on neurogenesis, and builds upon previous work (Peco 2012). The experimental results are, in general, sound. Some modelling is included, although this does not add significantly to the paper.

The highlight is genetic evidence that CDC25B has a cell-cycle-independent effect on differentiation. I find this to be an important point, given that there have been many correlative observations between cell cycle length and differentiation rate.

1) How faithfully is division mode measured?

In multiple places in the manuscript, claims are made regarding division mode (i.e. PP vs. NP vs. NN), e.g.

"symmetric neurogenic divisions require CDK interaction [with CDC25B]"

However, I am not 100% convinced with the measurement of division mode, given the present data.

Firstly, there is no direct live imaging/lineage tracing data to corroborate the measurement based on *Sox2*/tis21. Previous work (Saade et al. 2013) has shown that mitotic *Sox2*/tis21 expression predicts division mode in wildtype cells (although, as far as I am aware, there is a limited number of live cell tracks which directly back this up). However, it's not guaranteed that *Sox2*/tis21 remains a reliable division mode marker for the perturbations considered in this paper.

Secondly, this paper measures *Sox2*/tis21 in all progenitors; the original paper (Saade 2013), only in mitotic cells. Whilst Figure 4B does show that, on average, and at this time point, these measurements are similar, it means that the results quoted in this manuscript are less direct than in (Saade 2013), and there could be confounding effects due to the timing of reporter expression.

Therefore, I would suggest that either:i) the *Sox2*/tis21 measurements are complemented with a small number of direct lineage tracing experiments to confirm the main findings (e.g. that "symmetric neurogenic divisions require CDK interaction [with CDC25B]"), orii) the writing is changed to de-emphasize "division mode", and instead talk about overall "differentiation rate" (equivalently, the parameter "1 – γ" in the modelling section).

One reason that I think this distinction is important is that recent work in the chick neural tube [1] proposed that division mode statistics follow a binomial distribution i.e. daughter cells differentiate independently of one another. It is important to show whether the perturbations in this paper (e.g. CDC25B∆CDK) affect differentiation rate (γ) generally, or specifically generate a certain type of divisions (e.g. NP).

2) Model

Overall, I am not positive about the use of modelling in this paper. Whilst it is reassuring that the fairly simple model can fit a handful of datapoints on neuron fraction (Figure 6C), this is perhaps not all that interesting or impressive. Similar models have been generated previously [1].

What is interesting, is the claim:

"mathematical modelling reveals that cell cycle duration is not instrumental in controlling the mode of division."

However, I find the model suggested by Table 1 and Supplementary Information 3.2 a rather extreme scenario. To me, a more natural model would be where division modes are still probabilistic (with rates α_PP_, α_PN_, α_NN_), but now these rates are allowed to vary with cell cycle duration. It seems very difficult to rule out this more realistic model given the data presented.

[1] A Branching Process to Characterize the Dynamics of Stem Cell Differentiation, David Miguez, Scientific reports, 2015.

---

## [Author Response]

Reviewer #1:

[…] The study was well executed and the manuscript is well written. It would greatly strengthen the study if the authors could show the nature of the cell cycle-independent mechanism by which Cdc25B promotes neurogenic division, but this may be too much to ask.

We added the data obtained using the high resolution time‐lapse imaging technique we recently set up. We analysed in real time the behaviour of single neural progenitor nuclei during G2/M/G1 phases in the chicken developing neural tube and show that:

‐ During progenitor proliferation, nuclei basalward movements occurring in early G1 display two types of motion: slow (Ap nuclei) or fast (Bs nuclei) departure from the apical surface (Figure 8A‐D). To our knowledge it is the first time that this behavior is described in the spinal cord.

‐ The Mean Square Displacement (MSD) analyses suggest that Ap nuclei perform movements reminiscent of slow motion, and Bs nuclei exhibit directed movements (Figure 8E).

‐ CDC25B gain‐of‐function increases significantly the proportion of cells performing fast departure from the apical side without affecting the features of the movement (MSD, speed) (Figure 8D‐F). This indicates that the phosphatase controls the switch from slow to fast migration.

‐ This change occurs in proliferative progenitors indicating that it is upstream or independent of neurogenesis (see subsection “CDC25B promotes fast nuclei apical departure in early G1 independently of CDK interaction”, second paragraph).

‐ Importantly, the mutated form of CDC25B, inactive in the cell cycle, also induces this switch, reinforcing the idea that the phosphatase has a function outside its canonical role in the cell cycle and requiring a new substrate.

Reviewer #2:

This manuscript considers the role of CDC25B on neurogenesis, and builds upon previous work (Peco 2012). The experimental results are, in general, sound. Some modelling is included, although this does not add significantly to the paper.The highlight is genetic evidence that CDC25B has a cell-cycle-independent effect on differentiation. I find this to be an important point, given that there have been many correlative observations between cell cycle length and differentiation rate.1) How faithfully is division mode measured?[…] Therefore, I would suggest that either:i) the Sox2/tis21 measurements are complemented with a small number of direct lineage tracing experiments to confirm the main findings (e.g. that "symmetric neurogenic divisions require CDK interaction [with CDC25B]"), orii) the writing is changed to de-emphasize "division mode", and instead talk about overall "differentiation rate" (equivalently, the parameter "1 – γ" in the modelling section).One reason that I think this distinction is important is that recent work in the chick neural tube [1] proposed that division mode statistics follow a binomial distribution i.e. daughter cells differentiate independently of one another. It is important to show whether the perturbations in this paper (e.g. CDC25B∆CDK) affect differentiation rate (γ) generally, or specifically generate a certain type of divisions (e.g. NP).

We measured the division mode using another strategy besides Tis21/*Sox2* reporters. We used the Brainbow technique (Tozer et al., 2017).

2) ModelOverall, I am not positive about the use of modelling in this paper. Whilst it is reassuring that the fairly simple model can fit a handful of datapoints on neuron fraction (Figure 6C), this is perhaps not all that interesting or impressive. Similar models have been generated previously [1].

Our modeling work did not seek to be "impressive", and we apologize if it appeared to be so. We rather sought to be clear, explicit and robust about the hypotheses we made while interpreting the data. We would like to point out, in particular, that our model is not designed to "fit the data" by fine‐tuning free parameters, since it has no free parameters at all, a fact we consider as a strength. It was included here to check whether the modes of division (MoD) measured at HH17 were well in accordance with the neuronal fractions measuredat HH22, given the measured cell cycle length, and doing so with as few assumptions as possible. The good match obtained (at least in the ventral part) is presented for the following reasons: a) to underline our confidence in how the MoD were measured, since the simplest model shows that measured MoD explain quite nicely the evolution of neuronal fractions over 24 hours; b) as a tool to interpret the match obtained at the population scale in terms of what happens at the cell scale, namely to test whether cell cycle lengths associated with each mode of division are sufficient to fully determine the MoD, which we prove is not possible.

For these reasons, we think that our model is an important part of the paper. We also consider it important to supply the model it in full detail as supplementary information, for the sake of clarity and completeness for the readers.

To formulate our model, we started from standard textbook population dynamics (e.g., Segel LA and Edelstein‐Keshet L (2013) A Primer on Mathematical Models in Biology, Philadelphia, PA: Society for Industrial and Applied Mathematics) as did Saade et al., 2013. We only assumed that cell cycle length is stable over the time window of analyses (24 hours), and that mitoses are not synchronous, standard assumption in the domain. In any case, we thank the reviewer for drawing our attention to the reference (Míguez, 2015). We now refer to these models in the main text (subsection “Mathematical modelling reveals that cell cycle duration is not instrumental in controlling the mode of division”), and we added a short section to discuss this alternative in our Appendix (Preamble).

After examination of the model proposed in Míguez, 2015, we encounter a noteworthy difference between our model and the Míguez model (Míguez, 2015). Technically speaking, this difference is about how the limit to continuous time is considered. In Míguez, 2015, this limit is made considering all mitoses as synchronous and taking cell cycle length as tending to 0, (which is implicit in Equations 33‐35 of Supplementary Information of Míguez, 2015. For our model, we consider growth rate at the population scale, and then, we show that this population‐scale model corresponds to different interpretations at the cell scale. In particular, our model is a standard approximation for asynchronous mitotic cells with stable cell cycle length (that we recall in Figure 2—figure supplement 2). This corresponds to the most common observations for neuroprogenitors in the spinal cord, in the time window of our analysis (Molina and Pituello, 2017). Importantly, this difference between the two models yields different dynamics, especially when the balance between proliferation and differentiation of the progenitors is negative (i.e., in favor of differentiation, γ < 0). In the most extreme case (purely differentiating progenitors, γ = ‐1), our model still predicts dynamics correctly.

What is interesting, is the claim:"mathematical modelling reveals that cell cycle duration is not instrumental in controlling the mode of division."However, I find the model suggested by Table 1 and Supplementary Information 3.2 a rather extreme scenario. To me, a more natural model would be where division modes are still probabilistic (with rates α_PP_, α_PN_, α_NN_), but now these rates are allowed to vary with cell cycle duration. It seems very difficult to rule out this more realistic model given the data presented.[1] A Branching Process to Characterize the Dynamics of Stem Cell Differentiation, David Miguez, Scientific reports, 2015.

Our point was indeed to reject the extreme scenario, by showing that "the distribution of fates cannot be exclusively determined by differences in fate‐based cycle times". We added the term “extreme” in the main text accordingly (subsection “Mathematical modelling reveals that cell cycle duration is not instrumental in controlling the mode of division”, last paragraph).

We also noted that "It does not exclude that a given kind of fate (e.g., proliferative divisions PP) would require a longer time to be achieved than others, it excludes that such differences would suffice per se to explain the differences between the fractions of fates." The scenario suggested by the reviewer is interesting (that a given cell cycle duration could control the distribution of division modes). However, our measurement of cell cycle parameters shows that they are not significantly different between the control and the gain-of‐function conditions (Figure 2—figure supplement 1), so we did not consider this option.